# Non-invasive ultrasonic neuromodulation of the human nucleus accumbens impacts reward sensitivity

Siti N. Yaakub [1,2,11], John Eraifej [3,4,5,11], Nadège Bault [1,2,11], Mathilde Lojkiewiez[1,2], Elouan Bellec[1,2], Jamie Roberts[6], Noah S. Philip [7,8], Amir Puyan Divanbeighi Zand [4,5], Alexander L. Green [3,4], Matthew F. S. Rushworth [9,10,12] & Elsa F. Fouragnan [1,2,12] ✉

Precisely neuromodulating deep brain regions could bring transformative advancements in both neuroscience and treatment. We demonstrate that non-invasive transcranial ultrasound stimulation (TUS) can selectively modulate deep brain activity and affect learning and decision making, comparable to deep brain stimulation (DBS). We tested whether TUS could causally influence neural and behavioural responses by targeting the nucleus accumbens (NAcc) using a reinforcement learning task. Twenty-six healthy adults completed a within-subject TUS–fMRI experiment with three conditions: TUS to the NAcc, dorsal anterior cingulate cortex (dACC), or Sham. After TUS, participants performed a probabilistic learning task during fMRI. TUS-NAcc altered BOLD responses to reward expectation in the NAcc and surrounding areas. It also affected reward-related behaviours, including win–stay strategy use, learning rate following rewards, learning curves, and repetition rates of rewarded choices. DBS-NAcc perturbed the same features, confirming target engagement. These findings establish TUS as a viable approach for non-invasive deep-brain neuromodulation.

Brain-related health conditions affect one in four individuals worldwide[1]. Precise neuromodulation holds the potential to complement or even surpass current treatments by offering a targeted, non-pharmacological approach that directly modulates neural activity, providing personalised therapeutic treatment[2,3]. Harnessing ultrasound waves, typically employed in diagnostics, allows for the precise targeting of specific brain regions[4]. The method, called transcranial ultrasound stimulation (TUS), focuses ultrasound beams through the skull onto the brain, safely altering neural activity noninvasively[5–8]. TUS can reach deep brain regions with millimetric resolution and modulate discrete cell types within specific regions without affecting overlying brain areas[9–11].

Repetitive TUS produces neural changes that outlast the stimulation period itself. It has, therefore, been possible to design 'offline' TUS protocols with effects lasting hours and resembling early phase neuroplasticity and outlasting concurrent peripheral confounds[8,12].

[1]School of Psychology, Faculty of Health, University of Plymouth, Plymouth, UK. [2]Brain Research and Imaging Centre, Faculty of Health, University of Plymouth, Plymouth, UK. [3]Nuffield Department of Clinical Neurosciences, University of Oxford, Oxford, UK. [4]Nuffield Department of Surgical Sciences, University of Oxford, Oxford, UK. [5]Department of Neurosurgery, John Radcliffe Hospital, Oxford, UK. [6]Department of Clinical Measurement and Innovation, University Hospitals Plymouth NHS Trust, Plymouth, UK. [7]Department of Psychiatry and Human Behaviour, Warren Alpert Medical School of Brown University, Providence, RI, USA. [8]Center for Neurorestoration and Neurotechnology, VA Providence Healthcare System, Providence, RI, USA. [9]Wellcome Centre for Integrative Neuroimaging, University of Oxford, Oxford, UK. [10]Department of Experimental Psychology, University of Oxford, Oxford, UK. [11]These authors contributed equally: Siti N. Yaakub, John Eraifej, Nadège Bault. [12]These authors jointly supervised this work: Matthew F. S. Rushworth, Elsa F. Fouragnan. ✉e-mail: elsa.fouragnan@plymouth.ac.uk

Recent proof-of-concept studies using repetitive TUS at 10 Hz in non-human primates show changes in functional connectivity when TUS is targeted at cortical and subcortical regions[13]. These studies have been supplemented by others demonstrating that TUS also induces changes in cognition and behaviour[14–19]. In parallel, when careful precautions are taken to limit the transmission loss caused by the skull[20,21] recent studies in humans have demonstrated that TUS can effectively and precisely modulate activity and neurochemistry in deep parts of the cortex while participants are at rest[22]. Yet it remains to be determined whether applying TUS to deep subcortical structures during active cognitive engagement, such as during decision making and learning, can yield localised and specific changes in behaviour and neural function. If this is possible, not only does it open new avenues for testing causal hypotheses regarding human brain function, but it also raises the possibility of refined therapeutic applications.

Although studies in non-human primates have laid important groundwork by demonstrating that TUS can modulate behaviour through subcortical stimulation[16,23,24], translating these findings to humans presents non-trivial challenges. Structural differences in skull geometry and density affect ultrasound propagation and targeting precision, complicating direct comparisons. Moreover, human studies have so far operated under far more conservative acoustic energy limits due to regulatory constraints, most notably those set by the FDA for diagnostic ultrasound prior to the ITRUSST guidelines[25,26]. These constraints, which apply to energy deposition even before the ultrasound beam reaches the skull, have led to significantly lower dosing in human TUS protocols relative to those used in animal studies. As a result, demonstrating behavioural effects in humans under these conditions is not only technically challenging but also critically important for establishing translational relevance.

Here we focus on the nucleus accumbens (NAcc) in the ventral striatum, a deep subcortical region implicated in reward-guided learning in humans and other animals[27]. It is a key target of amygdala and mesolimbic dopaminergic projections[28,29]. Such inputs allow NAcc to guide behaviour based on reward outcomes expected after choices are made, and to facilitate learning and adaptation by signalling disparities between the predicted rewards and actual rewards. Midbrain dopamine neurons projecting onto NAcc carry both anticipated reward and prediction error signals, which can be described with classical reinforcement learning models[29–33]. In humans, functional magnetic resonance imaging (fMRI) studies align with this hypothesis, with evidence for reinforcement learning-based prediction error activity in midbrain[33] and NAcc[34–41].

Building on our own and others' previous work in humans and macaques, we sought to test whether TUS, using a system specifically designed to deliver stimulation deep in the human brain, could manipulate reward-related activity in NAcc. We then examined whether neural changes were associated with a change at a behavioural level. We were, therefore, guided by studies that have carried out causal manipulations of NAcc in macaques[42–47], emphasising the importance of both reward-guided, as opposed to loss-guided, aspects of behaviour, and stochastic rather than deterministic reward schedules for investigating NAcc. In aggregate, these studies emphasise the importance of stochastic rather than deterministic reward schedules for investigating the NAcc and that NAcc interventions in primates specifically affect reward-guided, as opposed to loss-guided, aspects of behaviour. We therefore used a related task in the current investigation and took special care to assess the degree to which behavioural changes were specific to reward-guided aspects of learning and decision making.

To establish the specificity of any effects found after NAcc stimulation, we included the dorsal anterior cingulate cortex (dACC) as an active control site. The dACC and NAcc are sometimes co-active and aspects of their roles in cognition are related, but crucially, there are also important differences[14,48–50] and if any impact of the dACC

intervention occurred, it was expected to manifest differently. Including the dACC, therefore, allowed us to test whether any observed behavioural or neural effects were specific to NAcc stimulation, as opposed to a more general response to TUS.

Whilst previous work has now demonstrated behavioural and neurophysiological modulation after TUS, it remains a novel stimulation strategy, particularly in humans. This is in contrast to electrical stimulation using deep brain stimulation (DBS) electrodes, which has become the standard of care for a number of neurological conditions, whilst also being used to treat psychiatric conditions and chronic pain[51]. The widespread clinical application of DBS has allowed for concurrent examination of micro, meso and macroscale circuit alterations that are associated with symptom response and behavioural modulation[52]. Given this knowledge of DBS, we therefore include DBS of the NAcc as a control stimulation method to probe possible mechanisms of action of TUS by directly comparing TUS effects with DBS effects.

In the current study, we took care to employ an especially cautious set of TUS parameters when testing human participants. We were therefore open to the possibility that changes in one of these parameters might lead to a facilitative or a disruptive effect, even while retaining the expectation that any effects would be specific to reward-guided learning and decision making. Our hypothesis was that non-invasive TUS could be used to causally modulate behaviour by selectively targeting the NAcc in humans. We predicted that stimulating the NAcc would specifically alter reward-guided learning, consistent with its role in processing reward prediction errors. In contrast, we did not expect similar behavioural effects following stimulation of the dACC, based on previous investigations of dACC stimulation in non-human primates and the activity patterns that have been reported in this region in both humans and non-human primates[14,48–50]. Our previous findings using magnetic resonance spectroscopy also showed minimal changes in GABA concentrations following TUS to the dACC, possibly due to the small focal area stimulated within a large and functionally heterogeneous Brodmann area. This distinction between regions enabled a stringent test of neural specificity.

By stimulating the NAcc during a probabilistic reversal learning task, known to robustly activate this region, we sought to demonstrate that TUS can drive specific, behaviourally meaningful modulation in a deep subcortical area in humans. This provides an important tool for causal manipulation in human neuroscience, enabling a better understanding of the functional roles of circumscribed brain regions. Such evidence would also represent a critical step toward establishing TUS as a targeted, nonpharmacological intervention for neuropsychiatric conditions.

## Results

### Transcranial ultrasound stimulation of the nucleus accumbens in humans

26 healthy adults participated in a within-subject repeated measures design with four visits, including an initial screening and MRI acquisition for planning TUS (session 1). Each of the subsequent three visits included TUS targeted at NAcc (TUS-NAcc), dACC (TUS-dACC), or a no sonication (Sham) condition. Visits were spaced at least one week apart and administered in a counterbalanced order across participants (sessions 2–4; Fig. 1a). During each of these sessions, participants first received TUS (or Sham) and were then placed in an MRI scanner and engaged in a probabilistic reversal learning task. Recordings of neural activity began ~10 min after the end of TUS application, when any potential auditory effects of the stimulation had dissipated.

Four task blocks were presented on average ~15, 28, 35, and 48 min after TUS (Fig. 1a). Through reward feedback, participants learned to choose the symbol with the highest probability of reward[14,34,35]. We employed the same task in previous work in humans[34,35] and macaques[14,15]. After 25 trials, the high reward probability was assigned to a different symbol, prompting participants to adapt (i.e. a 'reversal'

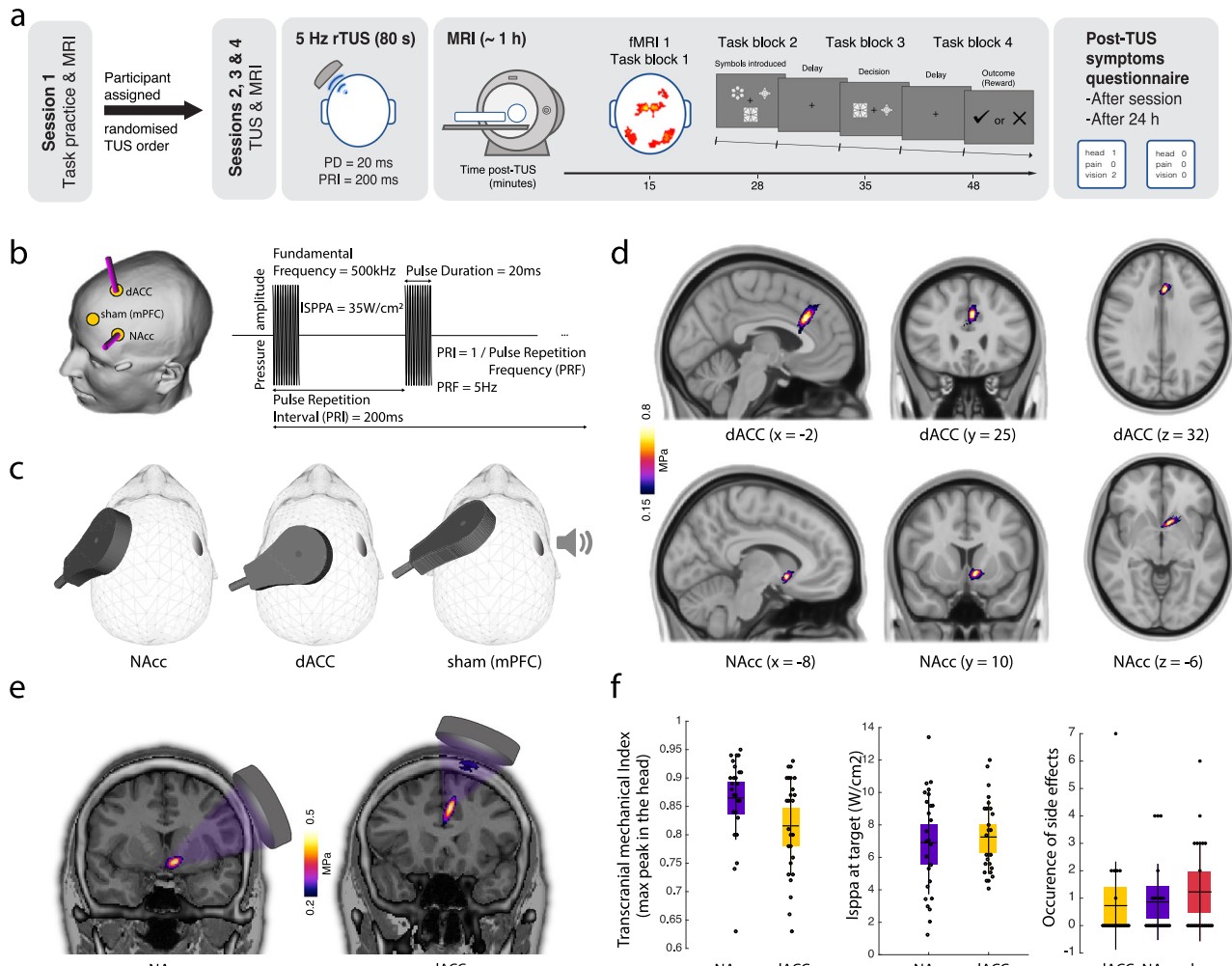

**Fig. 1 | TUS intervention. a** 26 participants attended four sessions. During the first, participants practiced a short version of the task and underwent anatomical MRI scans, after being assigned a randomised TUS condition order for the subsequent three TUS-MRI sessions (sessions 2–4). During these, participants underwent TUS targeted at the region determined by the condition order. This was followed by ~1 h of MRI during which they played the probabilistic learning task. At the start of each block, participants saw three symbols that would appear in that block. On each trial, participants saw two of the three symbols and were required to indicate the symbol that was most likely to lead to a reward. After a delay, they were presented with either a tick (reward), or a cross (no reward). Participants were presented with 320 trials divided into four task blocks. Every 25 trials, a reversal happened, forcing participants to relearn cue–reward contingencies. At the end of each session, participants filled in a questionnaire about any symptoms related to TUS. This was repeated at least 24 h after each session. **b** Target trajectories for neuronavigation

and TUS: 80 s of 5-Hz repetitive TUS (pulse duration (PD) = 20 ms and pulse repetition interval (PRI) = 200 ms). **c** Visualisation of the positioning of the transducer on the head and of the bone conductive headphones. **d** Averaged pressure maps across all participants restricted to each focal volume (as defined by −6 dB). The top row represents the average pressure map (FSLeyes NIH fire colour) across all individuals for the dACC while the bottom row represents the average pressure map for the NAcc (*n* = 26). **e** Example of individual acoustic simulations for both brain targets. **f** Transcranial mechanical index at the maximum peak in the head, usually in the skull (left panel), $I_{SPPA}$ at the intended brain target coordinate (middle panel), and occurrence of side effects after TUS–fMRI of the two TUS targets (right panel) (*n* = 26). Box plots show the mean and the standard error (bounds of the box). Data from each individual participant are presented as small black circles. Source data are provided as a Source Data file.

in reward contingencies occurred). Because the choice–reward associations were stochastic as opposed to deterministic, the task was similar to one previously found to be sensitive to NAcc intervention in macaques[47].

Figure 1 summarises the TUS approach, personalised planning, and results of acoustic simulation of TUS (Supplementary Tables 1 and 2). Targeting the left NAcc and dACC was determined based on Montreal Neurological Institute (MNI) coordinates, individually adjusted using T1-weighted MRI scans during target and transducer placement planning and acoustic simulation (Fig. 1b–d). Repetitive TUS comprised a 5 Hz-patterned protocol with a 10% duty cycle applied for 80 s. We planned the TUS intervention on each individual participant to maximise target engagement and minimise losses through the skull while ensuring safety prior to conducting the

three TUS sessions. This was achieved by positioning the transducer on each individual skull model while assessing TUS trajectory and in situ intensity and pressure. Importantly, we used a bespoke NeuroFUS system with a deep steering range (27.3 and 82.6 mm) making it possible to reach human NAcc ('Methods'). Although this was planned using a T1weighted-informed personalised pseudo-computerize tomography (CT)[21], other methods may achieve higher precision, for example using an ultrashort echo time image[53] or CT scans.

The spatial peak pulse average intensity ($I_{SPPA}$) in water was maintained at 35 W/cm² across participants[22,54]. In the sham TUS, no stimulation was delivered. Instead, participants heard a sound mimicking the TUS protocol sound, played via bone conduction headphones (Fig. 1c). Post-study verbal prompts revealed no discernible differences in perception between sessions. Figure 1d displays

averaged pressure maps across individuals for each target region while Fig. 1e presents a single participant. The acoustic simulation parameters and output for all study participants can be found in Supplementary Tables 1 and 2. Both NAcc and dACC were associated with similar maximum transcranial mechanical indices (MItc), $I_{SPPA}$ at focus (int eh region of the target), and occurrence of side effects (Fig. 1f; 'Methods'). We took care to remain within guidelines for human ultrasound exposure as defined by ITRUSST[26]. The results of our acoustic simulations predominantly indicated a maximum skull temperature rise below 2 °C. In cases where this threshold was exceeded, we calculated the Cumulative Equivalent Minutes at 43 °C (CEM43), a metric reflecting both duration and intensity of heating relative to 43 °C, the critical threshold for thermal cell damage. We ensured CEM43 values remained well below 0.25[26]. In our study, CEM43 was always below 0.1.

## TUS-NAcc induces specific changes in reward-related behaviours

During the task, participants' responses were probabilistic and aligned with the principles of a reinforcement learning mechanism. Model comparison revealed that a two learning rates model (2LRs)—for learning from rewarded and non-rewarded outcomes—fitted behaviours better than a simple Rescorla–Wagner (RW) model ($BIC_{RW} = 385.78$, $BIC_{2LRs} = 293.48$).

One of our main goals was to determine whether TUS-NAcc induces specific changes in reward-related behaviours in the hour following TUS. Given the importance ascribed to NAcc in reward-guided behaviour in macaque lesion studies[43–47], we focused on win–stay behaviours and ran a regression analysis across blocks, participants, and conditions ('Methods', Analysis A; Supplementary Table 4). This revealed a main effect of condition (ANOVA: $F_{2,75} = 3.3$, $p = 0.042$). Post-hoc t-tests revealed a stronger relationship between reward and subsequent win–stay behaviour after TUS-NAcc compared to Sham ($t_{25} = -2.75$, $p = 0.011$ Bonferroni corrected, Cohen $D = 0.53$, full results in Supplementary Table 3) and compared to TUS-dACC ($t_{25} = -3.52$, $p = 0.001$ Bonferroni corrected, Cohen $D = 0.69$). There was no significant difference between TUS-dACC and Sham ($t_{25} = -0.57$, $p = 0.573$, Cohen $D = 0.11$; Fig. 2a).

To test the temporal dynamics of TUS effects post-sonication, we examined the difference between TUS-NAcc and Sham for each of the four testing blocks collected approximately 15-, 28-, 35-, and 48-min post-TUS (averaged across participants and conditions). Similar TUS-NAcc effects were apparent at all times and the differences compared to Sham were even statistically significant at the block level at 28- and 35-min post-TUS. There was, however, no statistically significant difference between dACC and Sham at any time point (Fig. 2b; post-hoc tests in Supplementary Table 3).

Having found an increase in win–stay behaviours, we then looked at the learning curves across trials during a reversal period in the blocks identified in the previous analysis as demonstrating especially clear significant effects. We computed the rate of choice of the high probability option and found that participants were more likely to select the high probability option at the end of a reversal period after TUS-NAcc, compared to TUS-dACC and Sham (mixed-effect model: $t_{5924} = -2.74$, $p = 0.006$; Fig. 2c; Supplementary Table 4). This was particularly true for the first reversal ($t_{1478} = -3.97$, $p = 0.0007$; Fig. 2d; Supplementary Table 4).

To further characterise the behavioural effects of TUS, we conducted an additional analysis of trials involving two low-probability options. This revealed increased switching after TUS-dACC, consistent with disrupted counterfactual evaluation[14] (see Supplementary Fig. 1 and Supplementary Results). We also examined post-learning accuracy, perseveration errors, and post-error adjustments. These exploratory analyses did not reveal statistically significant differences between conditions. Specifically, post-learning accuracy showed no

significant main effect of TUS (mixed-effects model [PostLearning_Acc ~1 + TUS_session + (1 | sub)]: $t_{75} = 1.27$, $p = 0.209$), though we observed a trend toward an effect in the NAcc condition (post-hoc t-test, NAcc vs. Sham: $p = 0.07$). Perseveration errors also did not differ significantly between TUS conditions (mixed-effects model [Perf_Err ~1 + TUS_session + (1 | sub)]: $t_{75} = 0.363$, $p = 0.717$), nor did post-error adjustment (mixed-effects model [Post_Err ~1 + TUS_session + (1 | sub)]: $t_{75} = 0.165$, $p = 0.869$). These analyses are reported for completeness and to guide future investigations.

To look at the influence of reward history on learning, we fitted reinforcement learning models to the behavioural data and repeated all previous analyses with our best fitting reinforcement learning model estimates (Supplementary Fig. 2; Supplementary Tables 5–7).

Having established that TUS-NAcc impacted choices associated with the high probability option, we tested whether, overall, participants exhibited higher learning rates after TUS-NAcc estimated from the reinforcement learning model. This is indeed what we found. Learning rates linked with positive feedback were higher after TUS-NAcc compared to TUS-dACC and Sham (ANOVA: $F_{2,75} = 5.68$, $p = 0.005$; post hoc t-tests: TUS-dACC vs. TUS-NAcc: $t_{25} = -2.46$, $p = 0.021$; Cohen $D = 0.48$, TUS-NAcc vs. Sham: $t_{25} = -3$, $p = 0.006$; Cohen $D = 0.59$; TUS-dACC vs. Sham: $t_{25} = 0.96$, $p = 0.34$, Fig. 2e, Supplementary Table 7). Learning rates associated with non-reward feedback were not different across conditions (Fig. 2f). Although ventral striatum (including NAcc) activity reflects both aversive stimuli as well as rewarding stimuli, interventions, such as lesions, of ventral striatum in macaques, only affect reward-based learning[44]. Ventral striatal activity is present in deterministic and stochastic learning situations, but, again, as in the present study, only the latter are affected by ventral striatal lesions[43].

In general, win–stay strategies are adaptive because, by definition, they lead to the repetition of previously successful choices. Such a strategy may not be adaptive, however, after reward is experienced for choosing an option with a low average rate of reward. We, therefore, also investigated the rate of win–stay strategy when participants made choices between two options with low average reward rates. We found a main effect of condition (ANOVA: $F_{2,75} = 3.75$, $p = 0.02$) on the rate of maladaptive choices. Participants did not repeat the same low probability choice on the subsequent trial when a high probability symbol was on the screen in the Sham ($t_{25} = -0.3$, $p = 0.76$) or TUS-dACC condition ($t_{25} = 0.85$, $p = 0.39$), but they did after TUS-NAcc ($t_{25} = 3.24$, $p = 0.003$; Fig. 2g, Supplementary Table 8).

## Task-related TUS change in brain activity

Our behavioural analyses revealed an overall increase in reward sensitivity, which we explored in our main fMRI analysis by investigating the parametric blood oxygen level dependent (BOLD) responses to both reward expectation and delivery. Our main hypothesis was that there would be a task-related change in BOLD in the region targeted with TUS, which we first investigated with a region of interest analysis (ROI; 'Methods' and Supplementary Fig. 3). This revealed a clear increase in the parametric response to reward expectation in the NAcc in the TUS-NAcc condition compared to Sham and TUS-dACC condition (ANOVA: $F_{2,75} = 7.15$, $p = 0.001$; post hoc t-tests: dACC-NAcc: $t_{25} = -3.92$, $p = 0.0006$; Cohen $D = 0.77$; NAcc-sham: $t_{25} = -2.38$, $p = 0.024$; Cohen $D = 0.47$; dACC-sham: $t_{25} = -1.34$, $p = 0.19$; Cohen $D = 0.26$. Fig. 3a, Supplementary Table 9). A similar analysis revealed an increased response to reward delivery in dACC across conditions (ANOVA: $F_{2,75} = 3.12$, $p = 0.049$), however, the t-tests revealed that the comparison between TUS-dACC and Sham was not significant ($t_{25} = -1.47$, $p = 0.15$; Fig. 3b), although the difference between TUS-dACC and TUS-NAcc was ($t_{25} = -2.90$, $p = 0.007$).

This ROI analysis was followed by a whole brain analysis, which allowed us to investigate the full extent of the neural activity difference between TUS conditions. This analysis confirmed increased reward

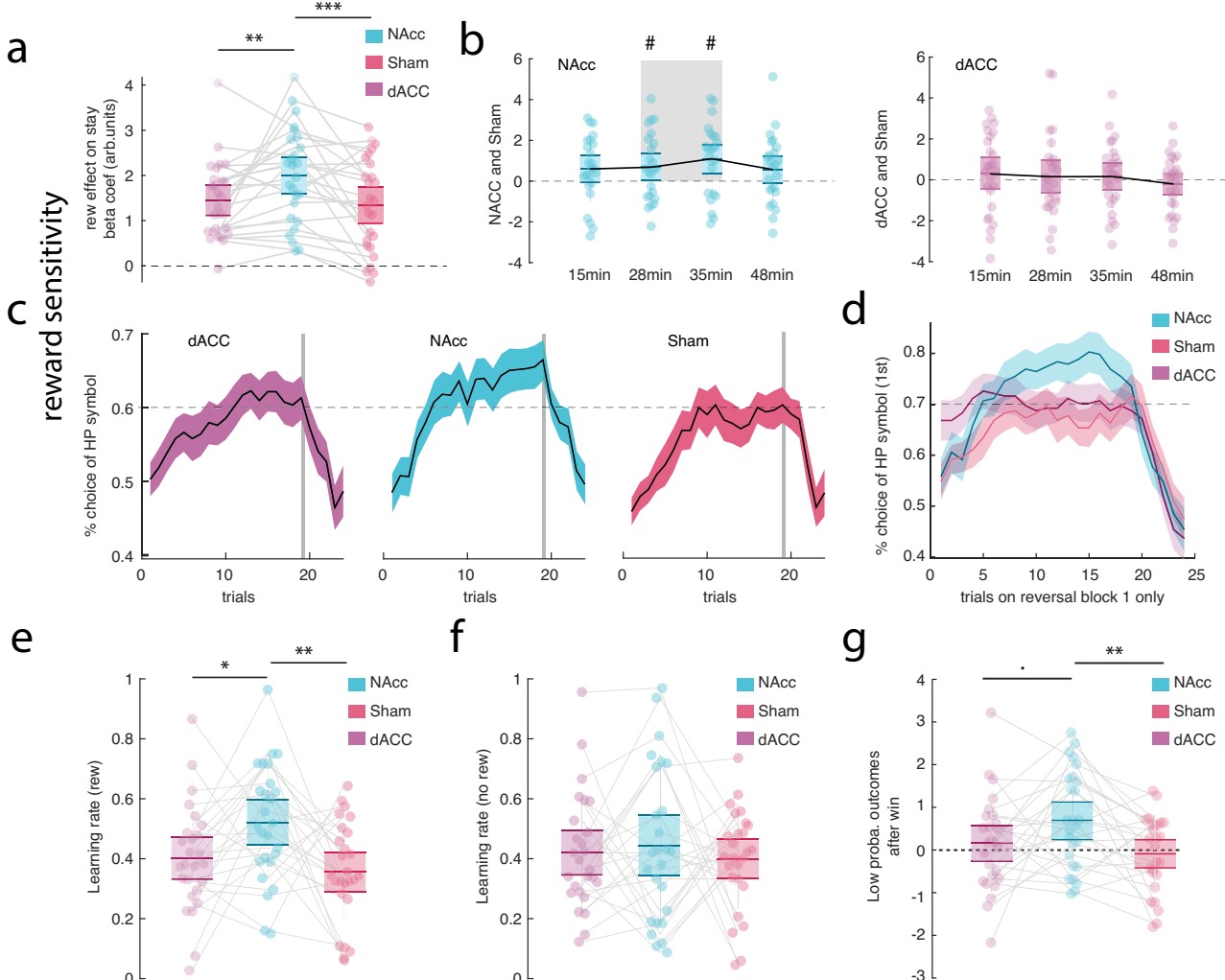

**Fig. 2 | Behavioural and model results. a** Win–stay analyses revealed an increase in the relationship between reward and subsequent win–stay behaviours after TUS-NAcc. Each dot represents an individual participant's mean beta estimate for the specified condition (*n* = 26). **b** Left panel: Time course of TUS-NAcc effects on reward-related behaviour, showing the difference between TUS-NAcc and Sham conditions for each of the four post-TUS testing blocks. The TUS-NAcc-induced reward-related changes were most prominent in the middle of the approximately 1-h post-TUS period. This effect was not observed in the right panel, which shows the comparison between TUS-dACC and Sham; no significant time window emerged in that contrast (*n* = 26). **c** Learning curves for the high-probability option across all reversal blocks. A 5-trial running average was applied to smooth trial-by-trial variability, which results in a slight shift in the apparent reversal point— appearing earlier than the actual reversal at trial 24 (trial 25 marks the start of the new reversal period). The shaded area represents the standard error of the mean.

**d** Same as (**c**) but showing only the first reversal block. The three conditions (TUS-NAcc, TUS-dACC, and Sham) are presented stacked and with transparency to allow for direct visual comparison. **e** Higher learning rates after reward feedback in the TUS-NAcc condition. Each dot represents an individual participant's estimated learning rate for the corresponding condition (*n* = 26). **f** We did not observe any change in learning rates after non-reward feedback after TUS-NAcc (*n* = 26). **g** Increase in rate of repetition of low probability options after reward showing maladaptive behaviour after TUS-NAcc condition compared to both TUS-dACC and Sham conditions (*n* = 26). n.s. non-significant. *p* < 0.1; *\**p* < 0.05; \*\**p* < 0.01; \*\*\**p* < 0.001; # significant windows. Exact *p* values are presented in the main text and in the supplementary material. Source data are provided as a Source Data file. **a**, **e**, **f**, **g** Statistical significance was determined using One-way ANOVA and two-sided t-tests. **b** Single two-sided t-tests were employed for each window. No multiple comparisons were applied.

---

expectation-related activity after TUS-NAcc compared to Sham, not only in NAcc itself, when it was the sonicated region, but also in the adjacent striatum, thalamus, amygdala, precuneus, and PCC (*Z* > 2.3; FDR corrected; Fig. 3c, left panel). The analysis also revealed increased reward expectation-related activity after TUS-NAcc compared to TUS-dACC, not just in NAcc but also in vmPFC, lOFC, insula, thalamus, putamen, precuneus, and PCC (*Z* > 2.3; FDR-corrected; Fig. 3c, middle panel). In dACC, however, there was no statistically significant difference for reward expectation between TUS-dACC and Sham conditions (Fig. 3c, right panel).

However, a whole-brain repeated measures ANOVA did identify an increased response to reward delivery after TUS-dACC compared to Sham in a distributed network adjacent to the sonicated region, in the

adjacent dACC and medial PFC (*Z* > 2.3; FDR-corrected; Fig. 3d, right panel). Although the ROI analysis had not revealed any reward-related activity differences precisely at the TUS-dACC target, the immediately adjacent dACC and medial prefrontal cortex did exhibit changes when the whole brain analysis was performed. The whole brain approach also revealed a stronger response to reward delivery after TUS-dACC compared to TUS-NAcc in a distributed network surrounding the dACC TUS focus (*Z* > 2.3; FDR-corrected; Fig. 3d, middle panel; compare dACC and NAcc bars in 3b). There was no statistically significant difference between reward delivery-related activity after TUS-NAcc and Sham (Fig. 3d, left panel).

On the day of stimulation and the following day, participants were prompted to report any adverse events by filling out a 4-point rating

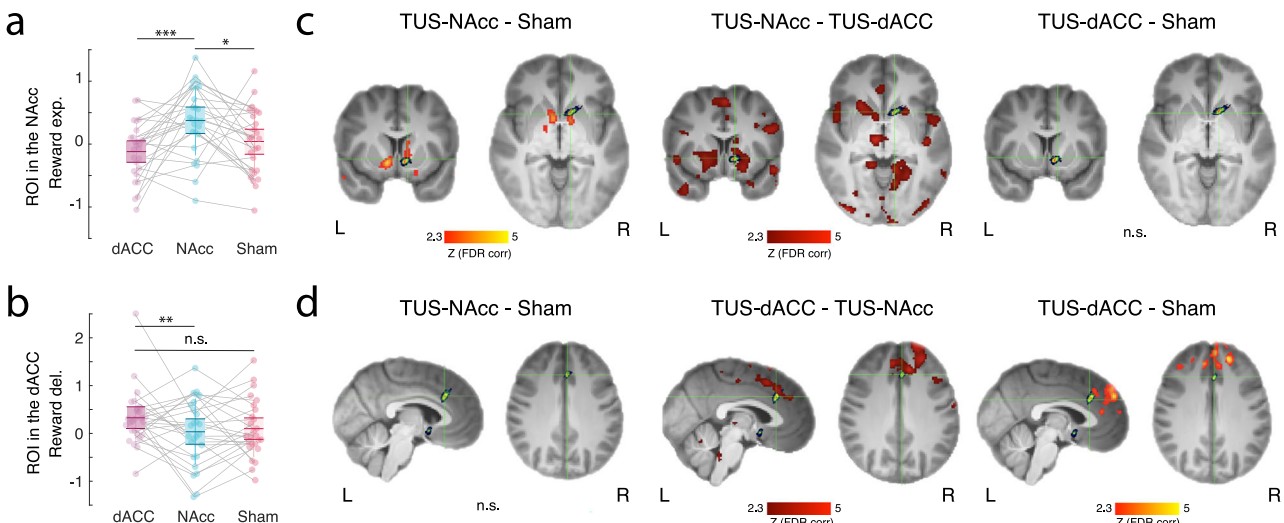

**Fig. 3 | ROI-based analysis and whole-brain differences. a** Enhancement of the parametric BOLD representation of reward expectation in the NAcc after TUS-NAcc compared to Sham and TUS-dACC (*n* = 26). **b** Although a main effect was observed in the dACC ROI for an increase in the BOLD representation of reward delivery after TUS-dACC, this effect was not significantly different from that seen in the Sham condition in the post-hoc test. TUS-dACC compared to TUS-NAcc was, however, significant (*n* = 26). **c** The same analysis at the whole-brain level revealed that TUS-NAcc not only increased the NAcc signalling of reward anticipation in the ROI but also in the adjacent striatum. Evidence of an even more distributed network of areas in which reward signalling was more prominent became apparent when TUS-NAcc was compared to TUS-dACC. However, consistent with (**b**), reward

expectation-related activity remained prominent in the NAcc and adjacent striatum. **d** Similarly, when contrasting the whole-brain maps between TUS-dACC to TUS-NAcc and Sham, we found strong evidence for BOLD changes in the dACC and medial frontal regions adjacent to the sonication site. This was not the case when contrasting TUS-NAcc to sham. FDR-corrected *Z* values of the differences between conditions are shown in red/orange and the averaged sonication site for dACC and NAcc are shown in green/blue. n.s. non-significant. *p* < 0.1; *p* < 0.05; **p* < 0.01; ***p* < 0.001; # significant windows. Exact *p* values are available in the main text as well as in the supplementary material. Source data are provided as a Source Data file. For **a**, **b** statistical significance was determined using One-way ANOVA and two-sided t-test. No multiple comparisons were applied.

scale questionnaire ('Methods')[55]. We found no difference between TUS and Sham conditions on the day of, and the day after, the sessions (Supplementary Fig. 4). However, upon qualitative inspection, it became evident that participants experienced various MRI-related issues, including sleepiness, neck pain, headache and blurry vision, which were attributed to the prolonged period of lying down and the lighting conditions inside the MR scanner.

### DBS-NAcc impacts the same reward-related behavioural indices as TUS-NAcc

So far, we have shown a correspondence between the effects of TUS-NAcc in humans and NAcc lesions in macaques[43,44]; both affect positive outcome-related behaviour on a probabilistic reward learning task. In a final experiment, we confirmed that direct high-frequency electrical DBS of bilateral human NAcc (Fig. 4a) also affected the same positive outcome-related behaviours in the probabilistic learning task. Three DBS-NAcc implanted patients, in a double-blinded, counterbalanced fashion, performed the probabilistic reversal learning task around 10 min after DBS was turned ON or OFF. The results show that DBS patients were more likely to select the high probability option when DBS was turned OFF than ON (Fig. 4b; mixed-effect model: $t_{452} = -5.65$, $p = 2.7\text{e-}08$), associated with a significant reduction in reward sensitivity (Fig. 4c; mixed-effect model: $t_4 = -5.06$ $p = 0.007$), which seemed to happen immediately after condition onset, suggesting a blunting of striatal response to reward sensitivity aligned with previous DBS-NAcc research[56].

Again, as with after human TUS-NAcc and macaque NAcc lesions[43,44], DBS-NAcc altered reward outcome-related behaviours. The same reward outcome-related behavioural indices were affected by both TUS-NAcc and DBS-NAcc; however, it is important to note that the direction of change for each index was opposite for TUS-NAcc and DBS-NAcc. It was also noticeable that the baseline reward sensitivity levels of the patients in the DBS-OFF state, all of whom had been treated with DBS for anorexia nervosa, (Fig. 4b, c) were higher

(Mann–Whitney test between DBS-OFF and healthy-TUS-Sham; U-stat = 67; $p = 0.045$) than those of healthy participants in SHAM. In fact, DBS-ON seemed to lower patient reward sensitivity to the healthy level (Mann–Whitney test between DBS OFF and healthy TUS-Sham; U-stat = 58; $p = 0.2$).

### Discussion

This study aimed to capitalise on the high spatial resolution of TUS and its capacity to reach deep regions in the brain to target the NAcc in humans in the context of probabilistic reversal learning. A total of 26 healthy participants were enroled in a within-subject repeated measures design experiment involving repetitive TUS and subsequent fMRI. After the application of 5 Hz patterned TUS for 80 s in a counterbalanced fashion to the NAcc (TUS-NAcc), the dACC (TUS-dACC), or no sonication (Sham), participants performed a probabilistic reversal learning task in the MRI scanner, which started on average 15 min post sonication, when any potential auditory or somatosensory effects of stimulation were dissipated. We used both direct measures of reward sensitivity and model-based estimates of the expected value associated with each potential choice stimulus. The models were also used to examine prediction errors when participants received feedback to indicate if the choice was rewarded or unrewarded in analyses of both behaviour and neural activity. We then compared these results with those of electrical DBS to the NAcc (DBS-NAcc) in a rare cohort of patients with electrodes in the NAcc.

With careful individualised TUS planning, using an estimate of each participant's skull image to achieve an optimised trajectory, it was possible to show that TUS-NAcc has neural effects that are most prominent in the region stimulated and that they are associated with changes in indices of behaviour that are similar to those emphasised in previous NAcc lesion studies[42–45] but also in DBS-NAcc as observed in this study. Indeed, we found significant alterations in reward-related behaviours, including alterations in the tendency to adopt a win–stay strategy, and a changed learning curve for the rewarding option.

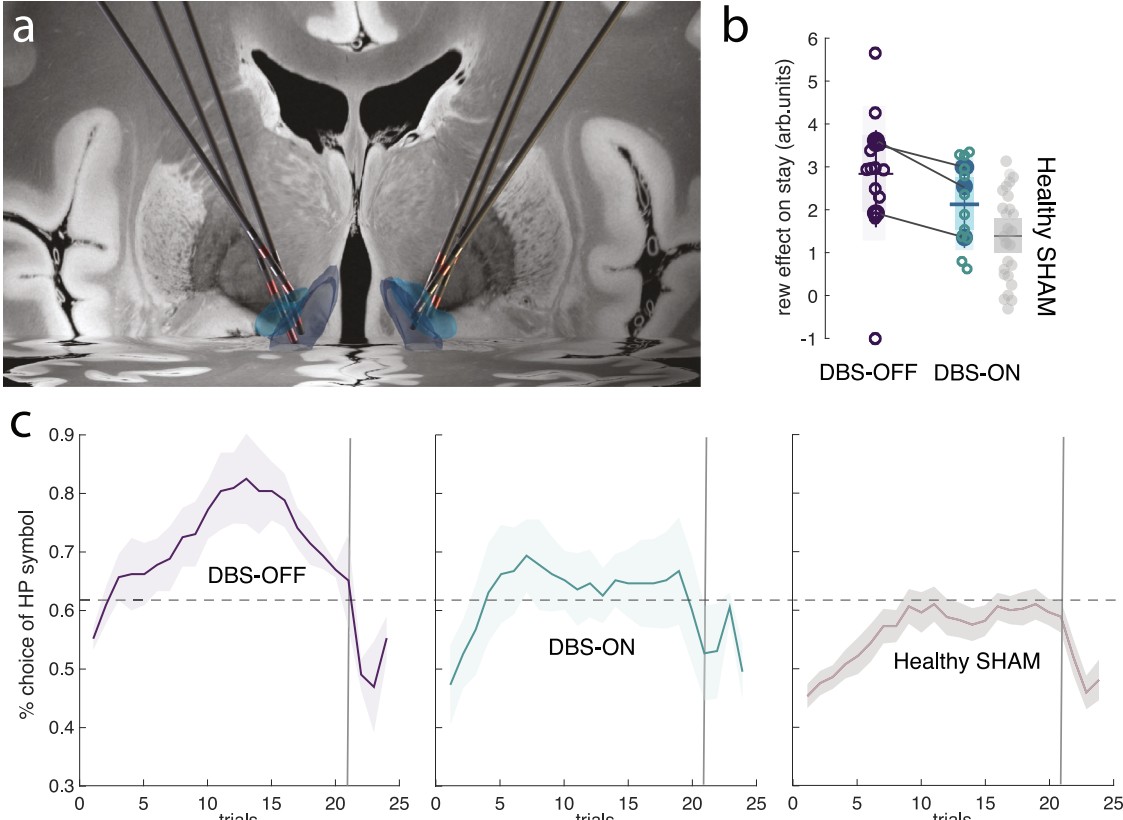

**Fig. 4 | DBS-NAcc investigation. a** Reconstruction of the DBS electrodes implanted in the bilateral NAcc to treat anorexia on a 3D coronal view of an MNI template brain with the NAcc core and shell presented in light blue and dark blue, respectively **b** Learning curves associated with the high probability option are presented across the whole session (i.e. four reversal periods). Each participant ($n = 3$) had a total of four sessions during ON and OFF DBS (so a total of 16 reversals for each condition). A 5-trial running average was applied to smooth trial-by-trial variability, which results in a slight shift in the apparent reversal point—appearing earlier than the actual reversal at trial 24 (trial 25 marks the start of the new reversal period). The

shaded area represents the standard error of the mean. The learning curve when the DBS is OFF is presented on the left, when DBS is ON, in the middle, and in healthy participants (from the Sham-TUS group), on the right. **c** Win–stay analyses revealed a decrease in the relationship between reward and subsequent win–stay behaviours after DBS-NAcc. Each dot represents an individual participant's mean beta estimate for a block and the specified condition ($n = 3$, four blocks each). The shaded area represents the standard deviation around the mean. The Sham-TUS is also shown for comparison ($n = 26$, healthy group). Source data are provided as a Source Data file.

While TUS offers high spatial precision, its effects can extend beyond the immediate target due to both anatomical proximity and the broader functional connectivity of the stimulated region. Although the protocol was carefully optimised for focal delivery, and behavioural effects were limited to one stimulation site, whole-brain analyses revealed more distributed neural activity when directly comparing the two active TUS conditions. Importantly, these effects appeared more spatially confined when each condition was compared to Sham, suggesting that the broader patterns reflect differential engagement of distinct networks rather than nonspecific or global activation.

This distinction is critical for interpreting the specificity of TUS effects. Localised behavioural outcomes, when paired with distributed neural changes, point toward functionally specific modulation within a larger interacting system. Indeed, evidence from the DBS literature suggests that stimulation of a single target can modulate activity of distinct neural networks in opposing directions, with both spatially[57] and temporally distinct dynamics[58,59]. Furthermore, small variations in electrode position can account for widespread differences in network engagement across a variety of neuroanatomical targets[60–63]. Therefore, while the spatial targeting of TUS remains a key strength, its influence should be interpreted not only in anatomical terms but also through the lens of circuit-level dynamics. Future work combining TUS with high-temporal-resolution methods or connectivity analyses could further elucidate

the causal relationships between focal stimulation and distributed neural responses.

Importantly, however, while the impact of TUS to any brain region is likely to be mediated through the connectional network that area has with the rest of the brain, it is equally important to remember that the connectional network of each area is unique[64]. This means that while two areas, A and B, might share connections with one another, and the effect of stimulation to either area might partly be mediated by a change in activity induced in the other area, it is also the case that areas A and B will always each have distinct aspects to their connectional networks; some aspects of area A's and area B's connectional networks will be distinct from one another. This point was underlined in the current study when the effects of TUS to a distinct brain region, dACC, were studied. Although there are some similarities in the activity patterns found in dACC and NAcc, as well as in other areas that project to both areas, such as the dopaminergic midbrain, there are also important differences[49]. When TUS was applied to dACC in the current study, it did not change the aspects of reversal task performance that were affected by NAcc TUS even though previous studies examining the effect of lesions, microstimulation, or TUS to dACC and adjacent perigenual anterior cingulate cortex have demonstrated other alterations in behaviour such as changes in the ability to track the value of counterfactual choices—choices that were not taken on the current trial but to which the animal might switch in the future[14]—and in learning from stochastic rewards and errors when an extended history

of outcomes[65] or uncertainty[48] must be taken into account, and changes in cost-benefit integration and motivation for task engagement[23,50,66–68] even though some of these effects are partly mediated via striatal regions adjacent to, and linked to, the NAcc[67,68].

Regarding the polarity difference between the TUS and DBS results, perhaps the most likely explanation is that the specific DBS and TUS parameters employed had opposing physiological effects. The specific TUS parameters used here are thought to be excitatory[22,69]. Conversely, though a simplification, high-frequency DBS is thought of as functionally inhibitory[70]. The behavioural effects we observed here are in keeping with the historical development of DBS for movement disorders, where high-frequency DBS was observed to have the same clinical effect as lesions in non-human primates[71,72]. Indeed, whilst high frequency subthalamic DBS is known to improve symptoms of Parkinson's disease with an associated reduction of pathological beta power[73], low frequency DBS to the same region increases beta power[74] with a correlated worsening of symptoms[75]. Similarly, low-frequency TUS has recently been reported to increase beta power in the same network[76]. Nonetheless, the simplicity and directness of translation from excitatory and inhibitory physiological changes to behavioural facilitation and impairment is unclear. Other factors might also be at play, namely the difference in baseline reward sensitivity seen between the healthy participants and the DBS cohort, all of whom had anorexia nervosa. It is known that reward sensitivity can be heightened in eating disorders[77,78] and particularly anorexia nervosa[79] compared to healthy controls, and that the ability of an individual to learn from rewarding stimuli is reduced after DBS-NAcc[80]. Therefore, the DBS-NAcc results reported here are in keeping with those previously described in the literature. It may be possible to identify TUS parameters that determine whether TUS exerts enhancing or disruptive effects. However, not only might the effects depend on many features of the TUS (intensity, frequency, and patterning of stimulation), but they might also depend on the anatomical structure of the brain region investigated and the baseline behavioural tendencies of participants. Given the great interest in the possibility of TUS-based therapies[2], such factors merit careful consideration before TUS is employed in patients whose conditions may include a range of changes in reward sensitivity, such as anorexia, substance use, bipolar disorder, or depression.

One limitation of the current study is the use of a constant free-field intensity for all participants, which may not effectively induce significant biological effects for some participants. Future research could focus on developing individualised neuromodulation protocols to maintain a constant in situ intensity for each participant. Additionally, customising TUS parameters based on an individual's baseline reward sensitivity could enhance efficacy. Such personalised approaches are important for both research and therapeutic applications. As the field advances and safety data accumulates, using intensities closer to those in animal models may also yield stronger TUS effects on cognition and brain activity. Finally, another limitation of the current study is the use of unilateral stimulation, which, although guided by current safety considerations for novel deep brain targets, does not allow us to fully assess potential lateralized effects in behaviour or neural response.

This study provides evidence that TUS provides a minimally invasive method that can, in humans, manipulate activity in a deep subcortical region to induce early phase neuroplasticity[8]. Many deep subcortical regions and subdivisions play crucial and specific roles in regulating fundamental behaviours and cognitive functions[81]. These regions are difficult to access using traditional neuromodulation techniques, and testing their causal roles in cognition in humans remains largely unexplored. Therefore, this study opens a potentially new and large space in which to examine hypotheses about human brain activity and its relationship with behaviours, with important lessons for future studies in neuropsychiatric conditions.

## Methods
### Participants
Twenty-six healthy volunteers (14 female; sex and gender aligned by self-report, with no participants identifying as non-binary) aged between 20 and 65 years (mean = 36.3, s.d. ± 13.3) participated in the study. Sex and gender information was collected based on self-report, and consent was obtained for reporting and sharing individual-level data. Participants were screened for contraindications to TUS and MRI[22], had no current diagnosis of neurological or psychiatric disorders, and were free of psychoactive medications at the time of the study. Written informed consent was obtained from all participants after experimental procedures were explained in full. The study was approved by the University of Plymouth Faculty of Health Staff Research Ethics and Integrity Committee (reference ID: 2487; date: 13/12/2021). Participants received £110 in total, including a performance bonus of £10 for completing all study sessions. Travel expenses were also reimbursed up to £10 per session. All healthy volunteer study sessions took place at the Brain Research & Imaging Centre in Plymouth, United Kingdom.

Three participants (3 females) aged between 31 and 61 (mean = 42, s.d. ± 13.5) treated with DBS of the NAcc for severe and enduring anorexia nervosa participated in the study (Supplementary table 10). Participants were screened for contraindications to turning off their DBS device for the duration of the experimental session, and written informed consent was obtained by the University of Oxford (reference ID: 12209; date: 05/03/2020). Participants were reimbursed for research-related travel expenses.

All 26 participants took part in all three TUS conditions (TUS-NAcc, TUS-dACC, and Sham) in a within-subjects design, with the order of conditions counterbalanced across participants.

### TUS protocol and procedure
**Study design.** The study design is summarised in Fig. 2a. Participants completed a behavioural practice and MRI only session followed by three TUS and MRI sessions, which were spaced at least one week apart and at the same time of day for each participant. During the behavioural practice and MRI session, participants familiarised themselves with the reversal learning task and underwent a short series of MRI scans, including a high-resolution T1-weighted MRI. This anatomical scan was used to derive a participant-specific head model for neuronavigation and acoustic simulations to plan TUS target and transducer placement for the subsequent TUS and MRI sessions. Participants were then assigned a randomised order of TUS conditions for their TUS and MRI sessions. During the TUS and MRI sessions, participants underwent 80 s of TUS followed by a series of MRI scans, including interleaved blocks of fMRI and MR spectroscopy scans, during which they performed the task (Fig. 1a). MR spectroscopy data will be the subject of another report. The TUS conditions were either active TUS applied to the left nucleus accumbens (TUS-NAcc), active TUS applied to the left dorsal anterior cingulate cortex (TUS-dACC), or Sham, where the transducer was placed as if to target the medial frontal cortex, but no ultrasound was delivered (Fig. 1b).

**Target location for ultrasound.** The left NAcc target was centred at MNI coordinates $x = -9$, $y = 11$, $z = -7$, and was identified based on an 80% probability atlas from the Harvard–Oxford Subcortical Structural Atlas supplied with FSL. The left dACC target was centred at MRI coordinates $x = -5$, $y = 24$, $z = 30$. During target and transducer placement planning, the targets were identified using an initial co-registration to MNI space and adjusted based on each individual's T1-weighted MRI.

**Acoustic simulations.** The output of our NeuroFUS transducer was previously measured with a hydrophone setup in a water tank (see Yaakub et al.[22] for details). We used the k-Wave Toolbox[82] (version 1.4) and custom scripts[21,22] in MATLAB (R2020b, MathWorks, Inc.) for our simulations, with an individual skull model estimated from each participant's T1-weighted MRI[21,22]. The codes to quantify the pseudo CT and to run the simulations can be found here[83,84]. The simulated transducer was modelled based on the physical properties of the NeuroFUS bespoke 4-elements transducer and optimised phases obtained using k-Plan (https://dispatch.k-plan.io). We set our simulation grid size to a 256 × 256 × 256 matrix centred on the midpoint between the transducer and focus with a grid spacing of 0.5 mm (i.e., 6 points per wavelength at 500 kHz).

**TUS protocol.** A bespoke CTX-500 NeuroFUS TPO system (Brainbox Ltd., Cardiff, UK) with a four-element annular transducer (diameter = 64 mm, central frequency = 500 kHz, and steering range between 27.3 and 82.6 mm) was used to deliver 5 Hz pulse repetition frequency repetitive TUS (pulse duration = 20 ms, pulse repetition interval = 200 ms, total duration = 80 s, total number of pulses = 400). The term bespoke reflects the custom specification of the transducer at the time of purchase in 2019, prior to the release of the standardised CTX-500 series. At that time, Sonic Concepts invited user-defined design parameters, including steering range. We selected a steering range (27.3–82.6 mm) tailored for targeting deep structures such as the human NAcc. As such, this transducer differs from later commercial CTX-500 units, which feature shallower, fixed ranges. This steering range ensured that it was possible to reach the NAcc in humans. The target free-field spatial-peak pulse-average intensity ($I_{SPPA}$) was kept constant at 35 W/cm² for all participants, which is the intensity before going through the skull and the soft tissue. Transcranial acoustic and thermal simulations (see 'Acoustic simulations' section for details) were performed both during planning and after each session to confirm that transcranial intensities remained within the limits of the ITRUSST safety guidelines for TUS[26].

To ensure good ultrasound transmission, we spread a layer of ultrasound transmission gel (Aquasonic 100, Parker Laboratories Inc.) into the hair where the transducer would be placed, making sure that air bubbles were smoothed or combed out. The head was not shaved. A 2 cm gel pad (Aquaflex, Parker Laboratories Inc.) was used between the transducer and the head. Neuronavigation was performed with Brainsight v 2.5 (Rogue Research Inc., Montréal, Québec, Canada) using the anatomical T1-weighted MR images from each participant. The focal depth read of Brainsight during each session was entered on the NeuroFUS TPO before stimulation was delivered. Once stimulation had begun, the trajectory was sampled with Brainsight and used in confirmatory acoustic simulations performed after each session. At the end of, and on the day after, each TUS session, participants were asked to report and elaborate on any adverse effects they thought were associated with TUS using a 4-point scale questionnaire with open-ended responses where they were encouraged to describe any experiences and whether they thought their experiences were related to the study procedures (provided in the Supplementary Material). Participants were blinded, while the experimenters were not.

Sham TUS was delivered in the same way as active TUS, except that the NeuroFUS TPO unit was turned off and no stimulation was delivered. A sound mimicking the sound produced by the envelope of the TUS protocol was played via bone conduction headphones (see ref. 22 for details) placed bilaterally approximately 2 cm posterior and superior to the temples. The same headphones were worn for all the conditions, including the NAcc and dACC active TUS conditions, but the sound was not played during those. We verbally prompted participants at the end of the study to disclose whether they had felt any difference between the sessions. Participants reported that they were unable to discern any differences between the sessions and had not suspected a Sham condition.

## DBS protocol and procedure

**Study design.** Participants completed a behavioural practice session followed by one session with DBS OFF and one session with DBS ON, the order of which were randomised and counterbalanced. One participant had DBS ON then OFF, and two participants had DBS OFF then ON. These sessions took place on the same day. During the behavioural practice, participants familiarised themselves with the reversal learning task. DBS was turned off for a 30 min washout period prior to the experimental sessions and both participants and assessors were blinded to condition allocation (DBS ON or DBS OFF). Participants then undertook four blocks of the reversal learning task in each condition, with 100 trials in each block. DBS ON refers to their DBS device being turned on at their usual therapeutic settings, for all participants this was at 130 Hz (Supplementary Table 10).

**DBS Surgery.** Participants were recruited into this study after completing a pilot study of DBS to the NAcc to treat severe and enduring anorexia nervosa (trial registration number: NCT01924598). Surgery was performed under general anaesthesia. A 2.7 mm twist drill craniotomy was made, and the electrode lead inserted bilaterally into the NAcc. All patients received intraoperative imaging to confirm electrode positioning was within the target, and the electrode was repositioned in real time if that was not the case. All electrode positions were confirmed using pre-operative MRI fused with post-operative CT with distal contact in NAcc and proximal contacts in the anterior limb of the internal capsule (ALIC). Target selection was based on anatomical/stereotactic references. All participants in this study had Medtronic 3387 electrodes with the Medtronic Activa RC model 37612, which is a constant voltage stimulator.

## MRI data acquisition and pre-processing

MRI scans were acquired on a Siemens MAGNETOM Prisma 3T scanner (VE11E, Siemens Healthineers, Erlangen, Germany) with a 32-channel head coil. The scans in this study included a T1-weighted MPRAGE sequence acquired in the sagittal plane (repetition time (TR) = 2100 ms, echo time (TE) = 2.26 ms, inversion time = 900 ms, flip angle (FA) = 8°, GRAPPA acceleration factor = 2, field of view = 256 × 256 mm, number of slices = 176, voxel size = 1 × 1 × 1 mm³), two GE-EPI fMRI scans during which participants performed the probabilistic reversal learning task lasting approximately 10-minutes each (acquisition plane tilted 30° clockwise from the line parallel to the AC–PC line, 1400 ms TR, 30 ms TE, 67° FA, 2.5 mm slice thickness, no slice gap, multi-band acceleration factor of 2, and 60 interleaved slices of 96 × 96 matrix size, giving a voxel size of 2.5 × 2.5 × 2.5 mm³), and field maps for fMRI distortion correction. FMRI pre-processing was performed using tools from the FMRIB Software Library v6.0 (FSL; www.fmrib.ox.ac.uk/fsl). Pre-processing included MCFLIRT motion correction, B0 inhomogeneity correction (effective EPI echo spacing = 0.49 ms, EPI TE = 30 ms, unwarp direction = −y, signal loss threshold = 10%), brain extraction, spatial smoothing (5 mm FWHM), and high pass filtering (0.01 Hz).

## Probabilistic reversal learning task

Healthy participants performed a probabilistic reversal learning task during four blocks of MR acquisitions (two of which were fMRI acquisitions). The task consisted of two runs of 100 trials each (presented during the fMRI scans) and two runs of 60 trials each (no fMRI), giving a total of 320 trials across four blocks. Three cues were presented per block, resulting in a total of 12 cues across the experiment (adapted from ref. 34). Additional stimuli included a tick to represent a reward, a cross to represent no reward, and a fixation cross.

In each task block, a different subset of three abstract symbols (e.g. A, B, and C) was randomly selected from the full set of 12 symbols. One of the three symbols would be associated with a 70% chance of obtaining a reward ('high' reward probability symbol) while the remaining two symbols were each associated with a 30% chance of obtaining a reward ('low' reward probability symbols). During each trial, participants were shown two of the three symbols and asked to select the symbol that they thought was associated with the highest probability of obtaining a reward. Participants were not informed of the exact reward probabilities assigned to each symbol but were instead asked to learn to choose the symbol that was more likely to lead to a reward through trial and error (i.e. by making use of the outcome of past decisions). Therefore, there was only one option associated with a high reward rate, while the other two were associated with lower reward rates. This procedure has a few advantages. First, because the best option is not presented on every trial, participants had to make different types of choices on different trials depending on which options were available. In addition, because for a third of the trials, participants had to choose between the two least rewarding symbols, even after they have learnt the cue–reward contingencies, this manipulation allowed us to achieve a more even distribution of reward and non-reward outcomes.

After making their decision, participants were shown either a tick to represent a reward on that trial or a cross to represent no reward. Participants were told that the number of rewarded trials would be counted across all the blocks and sessions, and that they would be awarded up to £10 as a performance bonus at the end of the study, depending how well they performed during the task. In the present task, outcomes were either reward or non-reward (i.e. reward omission); we use the term 'non-reward' throughout to reflect the absence of positive feedback without implying an explicit negative or punishing outcome.

To prevent participants from searching for non-existent patterns and to reduce cognitive load we presented the three possible pair combinations of the three symbols in a fixed order (i.e. AB, BC and CA). However, the presentation of the symbols on the left or right of the fixation cross each time was randomised. Participants were explicitly informed about this manipulation. After every 25 trials in each block, a 'reversal' would be introduced whereby the high reward probability was reassigned to a different symbol that was drawn out of the three symbols in total that were employed in each block. Participants would then have to learn to identify the 'new' high probability symbol out of the three. Participants were informed that reversals would happen several times during each block, but were not told of the exact frequency of reversals. The stochastic nature of the task and the reversals in choice–reward associations also ensured a need for constant assessment of each option's value and constant changes in choice selection.

Figure 1a shows the sequence of events for an example trial. At the start of each block of trials, the three abstract symbols selected for that block were shown on the screen for 5 s so that participants could familiarise themselves with the symbols they would see for the duration of that block. Each trial began with a fixation cross shown for a random delay of 1–1.2 s. This was followed by the presentation of two of the three symbols on either side of the fixation cross for 1.25 s, during which time participants were instructed to select one of the symbols by pressing a button. The fixation cross flickered for 100 ms after participants made their selection to indicate that their response was registered. The fixation cross was then shown again with a random delay of 1–1.2 s before the outcome of the trial, either a tick for a reward or a cross for no reward, was shown in the centre of the screen for 0.75 s. Trials in which participants failed to respond within 1.25 s were followed by a 'Lost trial' message shown in the middle of the screen in place of the tick or cross outcome.

## Stimuli display

The probabilistic reversal learning task was presented on an MRI-compatible LCD screen (BOLDscreen 32 AVI, 32-inch screen, resolution = 1920 × 1080, refresh rate = 120 Hz, Cambridge Research Systems Ltd., Rochester, Kent, UK) placed 1 m behind the MRI scanner, which participants viewed using a mirror attached to the head coil. The experiment was presented using the Presentation software (Neurobehavioural Systems Inc., Berkeley, CA, USA) run on a Windows machine. Responses were collected using an MRI-compatible fibre optic response pad (model: HHSC-2 × 4-C, Current Designs Inc., Philadelphia, PA, USA) placed in the participants' right hand. Participants selected the symbol on the right with their index finger and the symbol on the left with their middle finger.

## Modelling of behavioural data
### Models
**Model 1.** To create trial-wise estimates of the expected value (reward prediction) and prediction error, we first used a Rescorla–Wagner model, within the reinforcement learning framework, using each participants' behavioural choices and feedback. Specifically, the algorithm assigned each choice $i$ (for example selecting the symbol A) an expected value $V_A(t)$ which was updated via a prediction error, $\delta(t)$, as follows (1):

$$V_A(t+1) = V_A(t) + \alpha^* \delta(t), \tag{1}$$

where $\alpha$ is a learning rate that determines the influence of the prediction error on the updating of the symbol's expected value. The prediction error is calculated as (2):

$$\delta(t) = r(t) - V_A(t), \tag{2}$$

where $r(t)$ represents the outcome obtained on that trial (0 or 1). The expected values of the unselected stimulus (e.g., B) and the stimulus not shown on trial $t$ (e.g., C) were not updated.

**Model 2.** The main limitation of the classical Rescorla–Wagner model is the implied symmetry with which reward and non-reward feedback update a choice value estimate. This contradicts evidence that learning from reward and non-reward feedback has different effects on behaviour and decision-making[85]. We thus implemented an alternative asymmetric learning model which discriminates based on outcome valence, with two learning rates, one for reward and one for non-reward feedback[38,86].

With all models we used a SoftMax decision function in which, on each trial $t$, a stimulus choice probability (e.g. A) was given by (3):

$$P_A(i) = \sigma\big(\beta(V_A(t) - V_B(t))\big), \tag{3}$$

where $\sigma(z) = 1/(1 + e^{-z})$ is the logistic function, and $\beta$ represents the degree of choice stochasticity (i.e. the exploration/exploitation parameter). Choice probability of the unchosen stimulus (e.g. B) and of the stimulus not shown on trial t (e.g. C) were not updated.

The reinforcement learning model estimated expected value ($V$), prediction error ($\delta$), and separate learning rates ($\alpha^+$ for reward, $\alpha^-$ for non-reward) for each participant, block, and TUS condition. While expected value reflects the ongoing belief about how rewarding each option is, prediction error reflects the mismatch between expected and actual outcomes, and drives updating of value estimates. These model-derived variables were used to examine latent learning processes and were analysed separately from directly observed behaviours, such as win–stay strategies.

Estimates from these models were used in both behavioural regressions and as parametric modulators in fMRI analyses, enabling

us to dissociate stimulus-response learning from value-based decision signals.

**Model fitting and comparison of models.** We estimated parameters individually for each participant, block, and stimulation condition (TUS-NAcc, TUS-dACC, and Sham), including $\beta$ and the single or multiple learning rate $\alpha$. We initially determined reasonably good parameters by a grid search while applying the following parameter constraints: $\beta > 0$ and $0.1 < \alpha < 0.9$. The best parameters from the grid search were then used as starting points for a simplex optimisation procedure, which determined the final parameter estimates. As a goodness-of-fit measure, we used the log likelihood of the observed choices over all trials $T$ given the model and its parameters: $LL = \sum_{t=1}^{T} \ln[ft(y/\theta)]$, where $ft(y/\theta)$ denotes the probability of choice $y$ in trial $t$ given the model's parameter set $\theta$. Predicted choice probabilities were calculated based on 1000 simulations per parameter set (combinations of the free parameters), whereby in each simulation the model determined the choices used to update reward expectations (as opposed to observed choices). To obtain estimates of expected values and prediction errors, the two estimated participant-specific parameters were then re-entered into the reinforcement learning algorithm, this time based on participants' observed choices.

To determine the best fitting model, we performed classical model comparison. Specifically, for each model, we first estimated the subject-wise Bayesian Information Criterion (BIC) as follows (4):

$$BIC = -2 \log L + d \log n \qquad (4)$$

Here, the goodness of fit of a model ($-\log L$) is penalised by the complexity term ($d \log n$) where the number of free parameters in the model $d$ is scaled by the number of data points $n$ (i.e. trials). We then computed the sum of the subject-wise BIC for each model and compared the model-wise BIC estimates (lower estimates indicating better fit). The code to run these models and fitting procedure can be found here[87].

**Behavioural analysis**
Behavioural data were saved in standard ascii data files and were analysed using custom-written code in MATLAB (R2023a, MathWorks, Inc., https://uk.mathworks.com/). The relationships between Reward, Prediction Error, win–stay behaviours, and task structure were evaluated using linear regression. Statistical testing and post hoc analyses were performed using the default functions in MATLAB, namely *anova*, *fitglm*, *fitlme* and *ttest*. We ran two series of analyses, very similar in nature. The first one was run without estimates from the reinforcement learning model, simply the behaviours during the task. The second series used estimates from the reinforcement learning model.

**Analysis A.** The first analysis in the series was concerned with the relationship between reward and subsequent win–stay behaviours. We thus ran a regression model given by the Eq. (5):

$$WinStay(t+1) \sim Reward(t) + choiceStickiness(t) + isHPScreen(t+1) \qquad (5)$$

where WinStay$(t+1)$ is coded as +1 for Win–Stay, −1 for Win–Switch and 0 otherwise on trial $t+1$; Reward is coded as 1 if a reward is received and 0 if not on trial $t$; choiceStickiness is coded as 1 if choice has been repeated and 0 otherwise on trial $t$; and isHPscreen is coded as 1 if there was a high probability symbol on the screen and 0 otherwise. In essence, this is equivalent to looking at the proportion of win–stay behaviours while controlling for other features of the task and behaviour, such as any general tendency to choose repetition regardless of reward (choiceStickiness). We then subjected the

regression weights to an analysis of variance (ANOVA) such that (6):

$$regressionCoef \sim 1 + condition \qquad (6)$$

where condition is a categorical variable (TUS-dACC, TUS-NAcc or Sham). Further, three two-tailed two-sample t-tests were used for all possible TUS-dACC, TUS-NAcc and Sham pairs, applying Bonferroni correction. Cohen's $D$ was calculated for a paired-samples t-test by dividing the mean difference by the standard deviation of the difference.

**Analysis B.** The second analysis tested the difference between the regression weights from Analysis A for NAcc vs. Sham and dACC vs. Sham for each of the four blocks (2 fMRI and 2 tasks only). As these blocks were acquired at different times, this allowed us to plot the effect identified in Analysis A over time. We used two-tailed paired t-tests to check the significance of this difference for each block and considered $p < 0.05$ to be significant. We did not perform multiple comparisons as we used predefined hypotheses and tests.

**Analysis C.** The third analysis looked at the rate of choice of the high probability symbol, for the window significant in Analysis B, for the runs within a block, averaged across blocks, and then during the first run of a block, averaged across blocks.

We then repeated the Analyses A, B, and C with model estimates from the reinforcement learning model. In Analysis A, we used the prediction error at t instead of the reward at t. In Analysis C, we used the expected value associated with the high probability symbol instead of the choice rate associated with the high probability symbol and presented it for all four blocks.

**Task-based fMRI analysis**
FMRI data were pre-processed and analysed using FEAT (FMRI Expert Analysis Tool) Version 6.00, part of FSL (FMRIB's Software Library, www.fmrib.ox.ac.uk/fsl). Pre-processing included motion correction, B0 field inhomogeneity correction, brain extraction, spatial smoothing (5 mm FWHM) and high pass filtering (0.01 Hz). FMRI data were co-registered to the MNI standard space via a linear transform to the individual's high-resolution T1-weighted MRI and a non-linear transform to the MNI template.

**ROI analyses.** We hypothesised that we would find reward-related BOLD changes within the targeted areas of neuromodulation. For the NAcc site, we used the bilateral NAcc defined anatomically by the probabilistic Harvard–Oxford subcortical structural atlases. For the dACC site, we created a sphere around the maximum peak TUS intensity ($I_{SPPA}$ in situ) across participants extracted from our acoustic simulation (see Supplementary Fig. 4).

Statistical analyses of BOLD data were then performed using a fixed-effects approach within the framework of a GLM, as implemented in FSL (using the FEAT module). For each block, we ran a GLM in the form (7):

$$BOLD \sim UnmodFeedback + RewDel + RewExp + ReactionTime + Lost + counfounds \qquad (7)$$

where BOLD is a $T \times 1$ (T time samples) column vector containing the times series data for a given voxel; UnmodFeedback is an unmodulated regressor (all event amplitudes set to 1) locked at the time of outcome (that is, when the tick/cross appeared) as a boxcar regressor with a duration of 100 ms; RewDel is a simple categorical regressor for reward delivery (amplitudes set to +1 for rewarded outcomes) as a boxcar regressor with a duration of 100 ms; RewExp is a fully parametric regressor whose event amplitudes were modulated by the expected probabilistic reward associated with the chosen option,

locked at time of decision, as a boxcar regressor with a duration of 100 ms; ReactionTime is a boxcar regressor with a duration modulated by reaction time, locked at time of decision; and Lost is an unmodulated regressor for all lost trials, locked at time of decision. In addition, we included six nuisance regressors, one for each of the motion parameters (three rotations and three translations).

From each of the two reward-related regressors (RewExp and RewDel), we extracted beta coefficients from the NAcc and dACC by back-projecting the ROIs described above (Supplementary Fig. 4) from standard space into each individual's EPI (functional) space by applying the inverse transformations as estimated during registration. For the two reward-related regressors, we computed the average beta coefficients from all voxels in the back-projected ROIs and across participants to test the overall BOLD response profile of the ROIs as a function of both reward delivery and reward expectation.

**Whole-brain analyses.** We ran a single-group tripled t-test which corresponds to a repeated measures ANOVA with one fixed factor with three levels and one random factor. Fitting such a mixed effects model with ordinary least squares (OLS) (as implemented in FEAT) requires an assumption of compound symmetry. This is the state of equal variance and intra-subject correlations being equal. That is, Cov(scan1,scan2) = Cov(scan1,scan3) = Cov(scan2,scan3). For these whole-brain fMRI results, all images were thresholded given a one-sided t-test and subsequently FDR-corrected at $p < 0.05$.

### Reporting summary
Further information on research design is available in the Nature Portfolio Reporting Summary linked to this article.

## Data availability
The raw and processed MR data and acoustic simulation data generated in this study have been deposited in the Open Science Framework database under the CC-By Attribution 4.0 License: (https://osf.io/j34qz/, https://osf.io/w3mev/, and https://osf.io/vst9y/). Source data are also provided with this paper. Source data are provided with this paper.

## Code availability
FSL can be downloaded from https://fsl.fmrib.ox.ac.uk/fsl/fslwiki. The code to generate pseudo-CTs from T1-weighted images and the in-house acoustic simulation code based on k-Wave (Matlab) can both be found on GitHub (https://github.com/sitiny/mr-to-pct: https://doi.org/10.5281/zenodo.7110246 and https://github.com/sitiny/BRIC_TUS_Simulation_Tools: https://doi.org/10.5281/zenodo.8027240 respectively). K-wave can be downloaded from http://www.k-wave.org/. The reinforcement learning code is available on GitHub (https://github.com/efouragnan/RL_models: https://doi.org/10.5281/zenodo.16682779). For any further enquiries regarding the data, please contact Elsa Fouragnan. FSL can be downloaded from https://fsl.fmrib.ox.ac.uk/fsl/fslwiki. The code to generate pseudo-CTs from T1-weighted images and the in-house acoustic simulation code based on k-Wave (Matlab) can both be found on GitHub ((https://github.com/sitiny/mr-to-pct and https://github.com/sitiny/BRIC_TUS_Simulation_Tools, respectively). K-wave can be downloaded from http://www.k-wave.org/. For any further enquiries regarding the data, please contact Elsa F. Fouragnan.

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

## Acknowledgements
The authors thank Dr Maryann Noonan and Dr Marco Wittmann for fruitful discussions on the probabilistic task; research assistants Ema Darrieutort and Joshua Marquez for their work on the safety data, all study participants for taking part in the study, and the Brain Research & Imaging Centre (BRIC) MRI radiographers for their help with scanning. This research was supported by a UKRI Medical Research Council Future Leaders Fellowship, BBSRC, Neuromod + /ESPRC and ARIA grant (MR/T023007/1, BB/Y001494/1, EP/W035057/1 and SCNI-PR01-P15) (to E.F.F.). Scanning for this study was supported by the Brain Research & Imaging Centre (BRIC). Effort for N.S.P. was supported by the National Institutes of Mental Health (U01 MH123427) and US Dept of Veterans Affairs (I50 RX002864). The views expressed in this article are those of the authors and do not necessarily reflect the position or policy of the funders.

## Author contributions
S.N.Y. and E.F.F. conceived this research and designed the study. J.E., A.P.D.Z., and A.L.G. led the DBS studies and J.E. carried out the DBS experiments. N.S.P. advised on ultrasound safety of subcortical regions. J.R. advised on MRI acquisition. N.B. and M.R. advised on data quality and analysis. S.N.Y., E.B., M.L., N.B., and E.F.F. acquired data for the study. S.N.Y., J.E., N.B. and E.F.F. analysed the data. S.N.Y., J.E., N.B., M.R., and E.F.F. wrote the manuscript with input from all authors. All authors reviewed the final manuscript.

## Competing interests
E.F.F. is a consultant for Attune Neuroscience. N.S.P. is on the scientific advisory boards of Pulvinar Neuro and Grey Matter Neurosciences, and consultant to Motif Neurotech. All the other authors declare no competing interests.
