## [Transparent Peer Review file · Nature Communications]

Non-invasive Ultrasonic Neuromodulation of the Human Nucleus Accumbens Impacts Reward Sensitivity

Corresponding Author: Professor Elsa Fouragnan

Version 0:

Reviewer comments:

Reviewer #1

(Remarks to the Author)

Yaakub et al. used transcranial ultrasound stimulation (TUS) targeting the human nucleus accumbens (NAcc) and anterior cingulate cortex (ACC). They used a probabilistic reward learning task combined with fMRI and explored the effects of TUS on reward-guided learning. They also used invasive deep brain stimulation (DBS) for comparison. While the study does not offer significant new mechanistic insights into circuit or network mechanisms underlying adaptive or flexible learning, it can be a valuable study for clinical uses and provides a foundation for follow-up studies using this technique in human. Below, I list my major concerns followed by minor ones, which I hope can help the authors in improving the paper's quality and putting it in broader context.

1. My main difficulty with the paper is the lack of clarity about the purpose of the study. Is it merely replicating previous works to demonstrate the effectiveness of TUS in flexible cognition? Many explanations of results seemed post-hoc. The study targeted two brain regions while selectively focused on one without addressing much why TUS did not induce any effects on the dACC. Why was a task with more involvement of dACC not used? How does this study connect with TUS studies in primates that show impaired learning after sonication of ACC but not striatum, and other microstimulation studies in this area targeting both striatum and ACC finding non-similar effects, e.g.,:

Amemori et al., 2020. Microstimulation of primate neocortex targeting striosomes induces negative decision-making.

Banaie Boroujeni et al., 2022. Anterior cingulate cortex causally supports flexible learning under motivationally challenging and cognitively demanding conditions.

I believe the authors should discuss previous literature and the connections to their findings more thoroughly rather than focusing solely on supporting studies.

2. Another issue is unilateral brain stimulation without reporting cue lateralization or considering lateralization as a factor in the LME. Although the task involves higher-level cognitive processing, unilateral stimulation necessitates evaluating and reporting lateralized effects, if any.

3. Figures are in many cases unclear and lack details. For example, in Figure 3a, is each dot an individual subject or a learning block? It is unclear which period represents reward expectation versus delivery (this info is not provided either in the supplementary or methods sections). Figures need clearer explanations. also, some figure explanations are confusing, e.g.,:

"(A running average across 5 trials was used, which explained why the 207 reversal seems to happen at trial=19)." Why??

4. The DBS study is confusing. Why is the DBS-off condition different from the sham condition? Its connection to the overall study and contribution to the main findings is unclear. The explanation of authors for the opposite effects observed is highly speculative. The authors should integrate this part of the study more clearly.

5. The authors interpret non-reward outcomes as negative outcome, which is inaccurate. Negative outcome differs from no outcomes.

6. The authors need to explore TUS effects on various behavioral elements more thoroughly, including learning, post-learning accuracy, perseveration errors, and post-error behavioral adjustments.

7. As the study uses three stimuli, two with a 30% reward probability, in trials with two low-probability stimuli, there might be more ACC activation (as also shown by Hayden and Platt). I believe those trials need to be evaluated separately for the effect of TUS on the ACC, especially for BOLD signals.

8. The discussion is very brief and does not integrate the findings of the paper with the previous literature. The discussion needs to be better put together and discuss the connection of this paper to previous studies.

Minor comments:

The authors should report on reaction times for different conditions, as well as trials with and without high probability reward stimuli.

The text, in general, needs more edits, including technical ones. For example, on line 199: "we fitted the behavioural data to RL models and"; it should be the other way around (fitting models to the data).

It is unclear whether each session involved only 3 stimuli or 12. If 12, then clarify on what is a reversal in this context, as it is not currently clear in the methods section.

Potential off-target effects of TUS on other brain regions are not discussed despite broad neural activity changes noted.

In general, the text should clarify the RL model variables and their distinctions more effectively.

It is unclear whether participants were the same or different across various conditions. Should be explained.

Reviewer #2

(Remarks to the Author)

This is a beautifully written manuscript about a careful and well thought out set of experiments to demonstrate changes in behavior when sonicating the Nucleus Accumbens, as compared to control locations. It is an important study that should move the field forward. I have only a few comments that would provide clarity to the manuscript.

In Fig 1f, it is not clear what Mechanical Index is here. It is PNP derated by 0.3dB/cm/MHz and divided by the $\sqrt{f_0}$? Because this is the definition of mechanical index. Is it the PNP derated by some other derating factor for the skull and divided by the $\sqrt{f_0}$? This is then MI_{tc}. If it is MI_{tc}, then it should be labeled MI_{tc}.

Presumably, ISPPA at the focus and MI_{tc} would be related to each other. So, the maximum ISPPA at about 13 W/cm² would be related to the maximum MI_{tc}, which should be .91, but the figure has a maximum of .95. Can authors please explain.

Supp Table 1 - are columns 2-4 in free field? No, the methods says that the free field value was kept at 30 W/cm². So what are these three columns? Same for supplementary table 2.

page 16, the transducer is referred to as a bespoke transducer. According to miriam webster, bespoke means custom made. Is this transducer custom made? If it is custom made, there should be more details about it.

page 16, line 442, before going through the skull and presumably also the soft tissues.

Reviewer #3

(Remarks to the Author)

This study applied 80 s of low-intensity pulsed transcranial ultrasound to NAcc and dACC of 26 human subjects engaged in a probabilistic reward reversal task. In this task, the subjects had to learn which of three symbols is associated with the highest reward, before the reward contingencies switched.

There was a selective enhancement in parametric BOLD for both NAcc and dACC, relatively to sham.

The effects on behavior were assessed post-sonication, to eliminate potential artifacts that can be associated with the ultrasound.

There were substantial effects of NAcc stimulation (but not dACC stimulation) on the tendency to stay following a reward.

The probability of choosing the high-probability reward symbol was increased following NAcc stimulation.

There was also an appreciable effect on the learning rate following a reward.

The first two effects (tendency to stay following a reward and probability of choosing the high-probability reward symbol) were also observed in participants receiving NAcc DBS, although the ultrasound and DBS effects were of opposite polarity.

I find the study well executed. There is a sham condition, a control location (dACC), and the effects of ultrasound on behavior are evaluated first following the ultrasound offset, thus eliminating potential artifacts.

The demonstration of target engagement using fMRI BOLD is equally important.

The study is also well written.

The finding that effects of deep brain ultrasound stimulation on learning are observed even under relatively low (and safe) exposure levels qualifies the study, in my opinion, to be published.

I was only puzzled by the ultrasound and the DBS showing opposite effects. It is nonetheless possible, as the authors describe, that the two modalities do provide effects of opposite polarity under the stimuli used.

Version 1:

Reviewer comments:

Reviewer #1

(Remarks to the Author)

Authors have done a great job in addressing my comments. I have no further comment.

(Remarks on code availability)

Reviewer #2

(Remarks to the Author)

Thank you for the thoughtful responses. Just one comment —

Figure 1f still does not clarify whether it is MI or MI_{tc}. The response letter suggests it is MI_{tc}, but neither the graph label, nor the figure legend say this. In addition, the figure in the manuscript was not updated, as it does not match the figure in the response letter. But, even that figure does not clarify it is MI_{tc} in the figure label or in the caption.

(Remarks on code availability)

Reviewer #3

(Remarks to the Author)

The authors have addressed my suggestion.

(Remarks on code availability)

Responses to Reviewer #1:

Yaakub et al. used transcranial ultrasound stimulation (TUS) targeting the human nucleus accumbens (NAcc) and anterior cingulate cortex (ACC). They used a probabilistic reward learning task combined with fMRI and explored the effects of TUS on reward-guided learning. They also used invasive deep brain stimulation (DBS) for comparison. While the study does not offer significant new mechanistic insights into circuit or network mechanisms underlying adaptive or flexible learning, it can be a valuable study for clinical uses and provides a foundation for follow-up studies using this technique in human. Below, I list my major concerns followed by minor ones, which I hope can help the authors in improving the paper's quality and putting it in broader context.

We are grateful to the reviewer for acknowledging the potential of our study to inform clinical translation and to serve as a basis for subsequent TUS research in humans.

Major comments:

1) My main difficulty with the paper is the lack of clarity about the purpose of the study. Is it merely replicating previous works to demonstrate the effectiveness of TUS in flexible cognition? Many explanations of results seemed post-hoc. The study targeted two brain regions while selectively focused on one without addressing much why TUS did not induce any effects on the dACC. Why was a task with more involvement of dACC not used? How does this study connect with TUS studies in primates that show impaired learning after sonication of ACC but not striatum, and other microstimulation studies in this area targeting both striatum and ACC finding non-similar effects, e.g.,:

- Amemori et al., 2020. *Microstimulation of primate neocortex targeting striosomes induces negative decision-making.*
- Banaie Boroujeni et al., 2022. *Anterior cingulate cortex causally supports flexible learning under motivationally challenging and cognitively demanding conditions.*

I believe the authors should discuss previous literature and the connections to their findings more thoroughly rather than focusing solely on supporting studies.

We thank the reviewer for raising this important point. The primary objective of our study was to establish a clear causal link between activity in a deep subcortical human brain region and behaviour in humans, by using non-invasive TUS to directly modulate neural activity and observe corresponding behavioural changes.

While studies in non-human primates have previously demonstrated behavioural changes through stimulation of deep brain regions, there are several critical distinctions that make comparisons between non-human primates and human studies non-trivial. First, skull geometry and density differ substantially between species, influencing the transmission and focality of ultrasound energy. Most importantly, the acoustic energy levels used in human studies are subject to far more conservative constraints. Prior to the development of the ITRUSST safety guidelines (international consortium working together towards the safe and

effective application of transcranial focused ultrasound for neuromodulation), the only available regulatory benchmark was the FDA limit for diagnostic ultrasound, which imposed strict limits on energy deposition, particularly before the beam even reaches the skull. As a result, early human TUS studies, including ours, used stimulation intensities far below those typically employed in non-human primates research. This means that any observed effects in humans occurred under markedly lower energy conditions, highlighting both the sensitivity of the approach and the need for careful, species-specific translation of TUS protocols.

In this study, we targeted the nucleus accumbens (NAcc), using a well-validated task designed to probe reward-guided learning and its underlying neural mechanisms. Importantly, previous studies in non-human primate NAcc provide clear guidance about what aspects of the task might be affected by a circumscribed intervention in this area in humans. Importantly, however, the interventions undertaken in macaques are lesions and so while they suggest task features that should be focussed on, the previous studies do not provide strong constraints on the direction of effects to be expected in any task measure when stimulation is applied. Moreover, whether the direction of effect might be the same as that seen in the only other possible intervention in this area in humans – deep brain stimulation – was also unknown at the start of our study. These unknowns are a consequence of the fact that, to the best of our knowledge, this is the first demonstration of such targeted modulation in a deep brain region in humans. Probabilistic reversal learning was selected because it reliably engages the NAcc, given its role in reward processing, sensitivity to shifts in reward contingencies and its demand for flexible, adaptive decision-making. Using this focused approach, our results show that offline TUS to the human NAcc can alter reward sensitivity and influence learning, consistent with prior correlational findings in humans and causal evidence from non-human primates.

Regarding the role of the dACC, this region was included as an active control site, selected for its distinct functional profile relative to the NAcc and its established association with alternative functions. The inclusion of an active control site is now considered best practice in human TUS studies. In particular, this prevents the possibility of differential auditory or sensory confounds influencing results. Whilst offline protocols, where data are collected after the cessation of TUS, are less susceptible to these confounds, the inclusion of the dACC as an active control site ensures that the effects on reward-related processes are specific to the application of TUS to the NAcc.

In addition, the use of a reward-guided (rather than loss- or conflict-guided) task was intentional and aligned with our primary focus on NAcc function. While one might argue that the dACC could also be engaged during outcome processing, we viewed this as an opportunity to provide a more stringent test of regional specificity. Demonstrating selective effects in the NAcc, despite some potential overlap in decision-related functions, strengthens the case for dissociable neural and behavioral associations tied to specific deep brain structures, provided the underlying hypotheses are well defined.

While our whole-brain fMRI analysis revealed some neural modulation near the TUS-dACC site, behavioural effects specifically related to reward processing were non-significant. Effects related to choice behaviour are being explored in greater detail in a forthcoming, separate manuscript focused on the dACC's role in decision making and also answered in a separate question of the reviewer.

We acknowledge that this rationale could have been more clearly articulated in the introduction, where we intended to emphasize that our *a priori* goal was to reveal behavioral changes circumscribed to one neural target (NAcc) and not another (dACC), despite both being involved in aspects of decision making and despite carrying out a similar intervention associated with the same subjective experience for participants.

The introduction now reads:

Repetitive TUS produces neural changes that outlast the stimulation period itself. It has, therefore, been possible to design “offline” TUS protocols with effects lasting hours and resembling early phase neuroplasticity and outlasting concurrent peripheral confounds^{11,12}. Recent proof-of-concept studies using repetitive TUS at 10Hz in non-human primates show changes in functional connectivity when TUS is targeted at cortical and subcortical regions^{13–15}. These studies have been supplemented by others demonstrating that TUS also induces changes in cognition and behaviour^{14,16–20}. In parallel, when careful precautions are taken to limit the transmission loss caused by the skull^{21,22} recent studies in humans have demonstrated that TUS can effectively and precisely modulate activity and neurochemistry in deep parts of the cortex while participants are at rest²³. **Yet it remains to be determined whether applying TUS in humans to deep subcortical structures during active cognitive engagement, such as during decision making and learning, can yield circumscribed and specific changes in behaviour and neural function.** If this is possible, not only does it open new avenues for testing causal hypotheses regarding human brain function but it also raises the possibility of refined therapeutic applications^{24,25}.

Although studies in non-human primates have laid important groundwork by demonstrating that TUS can modulate behaviour through subcortical stimulation^{14,18,26}, translating these findings to humans presents non-trivial challenges. Structural differences in skull geometry and density affect ultrasound propagation and targeting precision, complicating direct comparisons. Moreover, human studies have so far operated under far more conservative acoustic energy limits due to regulatory constraints, most notably those set by the FDA for diagnostic ultrasound prior to the ITRUSST guidelines^{27,28}. These constraints, which apply to energy deposition even before the ultrasound beam reaches the skull, have led to significantly lower dosing in human TUS protocols relative to those used in animal studies. As a result, demonstrating behavioural effects in humans under these conditions is not only technically challenging but also critically important for establishing translational relevance.

Here we focus on the nucleus accumbens (NAcc) in the ventral striatum, a deep subcortical region implicated in reward-guided learning in humans and other animals. It is a key target of amygdala and mesolimbic dopaminergic projections^{29,30}. Such inputs allow NAcc to guide behaviour based on reward outcomes expected after choices are made, and to facilitate learning and adaptation by signalling disparities between the predicted rewards and actual rewards. Midbrain dopamine neurons projecting onto NAcc carry both anticipated reward and prediction error signals which can be described with classical reinforcement learning models^{30–34}. In humans, functional magnetic resonance imaging (fMRI) studies align with this hypothesis, with evidence for RL-based prediction error activity in midbrain³⁴ and NAcc^{35–42}.

Building on our own and others' previous work in humans and macaques, we sought to test whether TUS, using a system specifically designed to deliver stimulation deep in the human brain could manipulate reward-related activity in NAcc. We then examined whether neural changes were associated with change at a behavioural level. We were, therefore, guided by studies that have carried out causal manipulations of NAcc in macaques^{43–48}, emphasizing the importance of both reward-guided, as opposed to loss-guided, aspects of behaviour, and stochastic rather than deterministic reward schedules for investigating NAcc. In aggregate, these studies emphasize the importance of stochastic rather than deterministic reward schedules for investigating the NAcc and that NAcc interventions in primates specifically affect reward-guided, as opposed to loss-guided, aspects of behaviour. We therefore used a related task in the current investigation and took special care to assess the degree to which behavioural changes were specific to reward-guided aspects of learning and decision making.

To establish the specificity of any effects found after NAcc stimulation, we included the dorsal anterior cingulate cortex (dACC) as an active control site. The dACC and NAcc are sometimes co-active and aspects of their roles in cognition are related but, crucially, there are also important differences and if any impact of the dACC intervention occurred, it was expected to manifest differently. Including the dACC, therefore, allowed us to test whether any observed behavioural or neural effects were specific to NAcc stimulation, as opposed to a more general response to TUS.

Whilst previous work has now demonstrated behavioural and neurophysiological modulation after TUS, it remains a novel stimulation strategy, particularly in humans. This is in contrast to electrical stimulation using deep brain stimulation (DBS) electrodes which has become the standard of care for a number of neurological conditions whilst also being used to treat psychiatric conditions and chronic pain⁵¹. The widespread clinical application of DBS has allowed for concurrent examination of micro, meso and macroscale circuit alterations that are associated with symptom response and behavioural modulation⁵². Given this knowledge of DBS, we therefore include DBS of the NAcc as a control stimulation method to probe possible mechanisms of action of TUS by directly comparing TUS effects with DBS effects.

In the current study, we took care to employ an especially cautious set of TUS parameters when testing human participants. We were therefore open to the possibility that changes in one of these parameters might lead to a facilitative or a disruptive effect even while retaining the expectation that any effects would be specific to reward-guided learning and decision making. Our hypothesis was that non-invasive TUS could be used to causally modulate behaviour by selectively targeting the NAcc in humans. We predicted that stimulating the NAcc would specifically alter reward-guided learning, consistent with its role in processing reward prediction errors. In contrast, we did not expect similar behavioural effects following stimulation of the dACC, based on previous investigations of dACC stimulation in non-human primates and the activity patterns that have been reported in this region in both humans and non-human primates. Our previous findings using magnetic resonance spectroscopy also showed minimal changes in GABA concentrations following TUS to the dACC, possibly due to the small focal area stimulated within a large and functionally heterogeneous Brodmann area. This distinction between regions enabled a stringent test of neural specificity.

By stimulating the NAcc during a probabilistic reversal learning task, known to robustly activate this region, we sought to demonstrate for the first time that TUS can drive specific, behaviourally meaningful modulation in a deep subcortical area in humans. This provides an important tool for causal manipulation in human neuroscience, enabling a better understanding of the functional roles of circumscribed brain regions. Such evidence would also represent a critical step toward establishing TUS as a targeted, nonpharmacological intervention for neuropsychiatric conditions.

Additionally, we now include additional discussion in the manuscript to clarify how our findings complement primate studies showing dissociable effects of TUS and microstimulation across ACC and striatal regions (e.g., Amemori et al., 2020; Banaie Boroujeni et al., 2022), and to emphasize that the NAcc's involvement in reward-driven learning was specifically targeted here, while ACC-related effects under different motivational demands will be reported separately. The discussion now reads:

While TUS offers high spatial precision, its effects can extend beyond the immediate target due to both anatomical proximity and the broader functional connectivity of the stimulated region. Although the protocol was carefully optimized for focal delivery, and behavioural effects were limited to one stimulation site, whole-brain analyses revealed more distributed neural activity when directly comparing the two active TUS conditions. Importantly, these effects appeared more spatially confined when each condition was compared to Sham, suggesting that the broader patterns reflect differential engagement of distinct networks rather than nonspecific or global activation.

This distinction is critical for interpreting the specificity of TUS effects. Localized behavioural outcomes, when paired with distributed neural changes, point toward functionally specific modulation within a larger interacting system. Indeed, evidence from the DBS literature suggests that stimulation of a single target can modulate activity of distinct neural networks in opposing directions, with both spatially⁶⁰ and temporally distinct dynamics^{61,62}. Furthermore, small variations in electrode position

can account for widespread differences in network engagement across a variety of neuroanatomical targets^{63–66}. Therefore, while the spatial targeting of TUS remains a key strength, its influence should be interpreted not only in anatomical terms but also through the lens of circuit-level dynamics. Future work combining TUS with high-temporal-resolution methods or connectivity analyses could further elucidate the causal relationships between focal stimulation and distributed neural responses.

Importantly, however, while the impact of TUS to any brain region is likely to be mediated through the connectional network that that area has with the rest of the brain, it is equally important to remember that the connectional network of each area is unique⁶⁷. This means that while two areas, A and B, might share connections with one another, and the effect of stimulation to either area might partly be mediated by a change in activity induced in the other area, it is also the case that areas A and B will always each have distinct aspects to their connectional networks; some aspects of area A's and area B's connectional networks will be distinct from one another. This point was underlined in the current study when the effects of TUS to a distinct brain region, dACC, was studied. Although there are some similarities in the activity patterns found in dACC and NAcc, as well as in other areas that project to both areas such as the dopaminergic midbrain, there are also important differences⁵⁰. When TUS was applied to dACC in the current study, it did not change the aspects of reversal task performance that were affected by NAcc TUS even though previous studies examining the effect of lesions, microstimulation, or TUS to dACC and adjacent perigenual anterior cingulate cortex have demonstrated other alterations in behaviour such as changes in the ability to track the value of counterfactual choices – choices that were not taken on the current trial but to which the animal might switch in the future¹⁷ – and in learning from stochastic rewards and errors when an extended history of outcomes⁶⁸ or uncertainty⁴⁹ must be taken into account, and changes in cost-benefit integration and motivation for task engagement^{14,51,69–71} even though some of these effects are partly mediated via striatal regions adjacent to, and linked to, the NAcc^{70,71}.

2) Another issue is unilateral brain stimulation without reporting cue lateralization or considering lateralization as a factor in the LME. Although the task involves higher-level cognitive processing, unilateral stimulation necessitates evaluating and reporting lateralized effects, if any.

We appreciate the reviewer's point regarding the consideration of lateralized effects. However, our decision to apply unilateral stimulation was made deliberately and in accordance with current safety considerations for TUS, particularly for deep brain targets like the NAcc which had not been sonicated in humans at the time the study began.

Specifically, the NIH and international TUS safety guidelines advise caution with bilateral sonication in novel targets, emphasizing the importance of first establishing safety and efficacy in unilateral protocols. While bilateral stimulation is common in electrical DBS interventions for conditions such as eating disorders or addiction, TUS operates via fundamentally different mechanisms and does not yet share the same regulatory precedent or safety profile.

Our aim was to first demonstrate that unilateral TUS can safely and effectively modulate behavioural and neural processes in a targeted and specific manner, that could also provide an important reference for future studies in clinical populations. As such, we followed a conservative and scientifically justified approach.

However, we appreciate that lateralized effects are an important consideration, and we explored this by conducting ROI analyses comparing activity in each hemisphere. These analyses did not reveal any significant differences between left and right ROIs within each target region.

Nonetheless, we agree that future studies could investigate laterality more directly, especially as more safety data emerges. We have now added a note in the Discussion to clarify this rationale and to highlight the potential for follow-up work on hemispheric differences.

Extract from the discussion:

One limitation of the current study is the use of a constant free-field intensity for all participants, which may not effectively induce significant biological effects for some participants. Future research could focus on developing individualized neuromodulation protocols to maintain a constant in situ intensity for each participant. Additionally, customizing TUS parameters based on an individual's baseline reward sensitivity could enhance efficacy. Such personalized approaches are important for both research and therapeutic applications. As the field advances and safety data accumulates, using intensities closer to those in animal models may also yield stronger TUS effects on cognition and brain activity. **Finally, another limitation of the current study is the use of unilateral stimulation, which, although guided by current safety considerations for novel deep brain targets, does not allow us to fully assess potential lateralized effects in behaviour or neural response.**

3) Figures are in many cases unclear and lack details. For example, in Figure 3a, is each dot an individual subject or a learning block? It is unclear which period represents reward expectation versus delivery (this info is not provided either in the supplementary or methods sections). Figures need clearer explanations. also, some figure explanations are confusing, e.g.,: "(A running average across 5 trials was used, which explained why the 207 reversal seems to happen at trial=19)." Why??

We thank the reviewer for pointing this out. We agree that clear and well-annotated figures are essential for communicating our findings effectively.

In response, we have revised the figures and legends throughout the manuscript to ensure they are more intuitive and self-contained. Specifically for Figure 2, 3 and 4, we now clarify that each dot represents an individual participant's mean beta estimate for the specified condition. We have also updated the legend and axis labels to clearly distinguish reward expectation and reward delivery periods, and have added corresponding information to the Methods and Supplementary Materials for full transparency.

Regarding the reviewer's example: "(A running average across 5 trials was used, which explained why the reversal seems to happen at trial=19)."

We recognize that this explanation was confusing. We have now rephrased this sentence for clarity, noting that the running average smooths the learning curve, which causes the apparent reversal point to shift slightly earlier than the actual trial 24 switch (25 being a new reversal period). This has been clarified in the figure caption:

Fig.2. Behavioural and model results. **a** Win–stay analyses revealed an increase in the relationship between reward and subsequent win–stay behaviours after TUS-NAcc. **Each dot represents an individual participant's mean beta estimate for the specified condition.** **b** Left panel: Time course of TUS-NAcc effects on reward-related behaviour, showing the difference between TUS-NAcc and Sham conditions for each of the four post-TUS testing blocks. The TUS-NAcc-induced reward-related changes were most prominent in the middle of the approximately one-hour post-TUS period. This effect was not observed in the right panel, which shows the comparison between TUS-dACC and Sham; no significant time window emerged in that contrast. **c** Learning curves for the high-probability option across all reversal blocks. A 5-trial running average was applied to smooth trial-by-trial

variability, which results in a slight shift in the apparent reversal point—appearing earlier than the actual reversal at trial 24 (trial 25 marks the start of the new reversal period). The shaded area represents the standard error of the mean. **d** Same as **c** but showing only the first reversal block. The three conditions (TUS-NAcc, TUS-dACC, and Sham) are presented stacked and with transparency to allow for direct visual comparison. **e** Higher learning rates after reward feedback in the TUS-NAcc condition. Each dot represents an individual participant's estimated learning rate for the corresponding condition. **f** We did not observe any change in learning rates after non-reward feedback after TUS-NAcc. **g** Increase in rate of repetition of low probability options after reward showing maladaptive behaviour after TUS-NAcc condition compared to both TUS-dACC and Sham conditions. . $p < 0.1$; * $p < 0.05$; ** $p < 0.01$; *** $p < 0.001$; # significant windows.

We truly appreciate the reviewer's attention to these details, and we hope the revised figures and explanations now provide a much clearer and more accessible representation of the data.

4) The DBS study is confusing. Why is the DBS-off condition different from the sham condition? Its connection to the overall study and contribution to the main findings is unclear. The explanation of authors for the opposite effects observed is highly speculative. The authors should integrate this part of the study more clearly.

We thank the reviewer for the opportunity to clarify the relevance of the DBS component of this work.

Firstly, in order to integrate this part of the study more clearly, the rationale for including a DBS cohort in the study has been outlined in our introduction. We highlight that whilst TUS is becoming widely adopted as a means of neuromodulation both in neuroscientific studies and clinical pilot studies, it remains novel and relatively poorly understood. On the other hand, DBS is now established as the standard of care in a number of neurological conditions and its uptake has allowed for a much deeper understanding of the possible mechanisms of action that may explain observed results during electrical stimulation. For this reason, DBS was included as a control stimulation method to provide a means of probing possible mechanisms of action of TUS during our probabilistic reversal learning task. The following paragraph has now been included in the introduction:

Whilst previous work has now demonstrated behavioural and neurophysiological modulation after TUS, it remains a novel stimulation strategy, particularly in humans. This is in contrast to electrical stimulation using deep brain stimulation (DBS) electrodes which has become the standard of care for a number of neurological conditions whilst also being used to treat psychiatric conditions and chronic pain⁵². The widespread clinical application of DBS has allowed for concurrent examination of micro, meso and macroscale circuit alterations that are associated with symptom response and behavioural modulation⁵³. Given this knowledge of DBS, we therefore include DBS of the NAcc as a control stimulation method to probe possible mechanisms of action of TUS by directly comparing TUS effects with DBS effects.

The first paragraph of the discussion has also been revised:

This study aimed to capitalize on the high spatial resolution of TUS and its capacity to reach deep regions in the brain to target the NAcc in humans in the context of probabilistic reversal learning. A total of 26 healthy participants were enrolled in a within-subject repeated measures design experiment involving repetitive TUS and subsequent fMRI. After the application of 5Hz patterned TUS for 80 seconds in a counterbalanced fashion to the NAcc (TUS-NAcc), the dACC (TUS-dACC), or no sonication (Sham), participants performed a probabilistic reversal learning task in the MRI scanner which started on average 15min post sonication, when any potential auditory or somatosensory effects

of stimulation were dissipated. We used both direct measures of reward sensitivity and model-based estimates of the expected value associated with each potential choice stimulus. The models were also used to examine prediction errors when participants received feedback to indicate if the choice was rewarded or unrewarded in analyses of both behaviour and neural activity. We then compared these results with those of electrical deep brain stimulation to the NAcc (DBS-NAcc) in a rare cohort of patients with electrodes in the NAcc.

The relevance of DBS insights in the interpretation of TUS results has been demonstrated in a few discussion points. For example, the networked effects of DBS in clinical populations highlight that even with very precise anatomical targeting, with verification on post-operative CT, behavioural and connectivity effects can vary between people:

This distinction is critical for interpreting the specificity of TUS effects. Localized behavioral outcomes, when paired with distributed neural changes, point toward functionally specific modulation within a larger interacting system. Indeed evidence from the DBS literature suggests that stimulation of a single target can modulate activity of distinct neural networks in opposing directions, with both spatially⁶⁰ and temporally distinct dynamics^{61,62}. Furthermore, small variations in electrode position can account for widespread differences in network engagement across a variety of neuroanatomical targets⁶³⁻⁶⁶. Therefore, while the spatial targeting of TUS remains a key strength, its influence should be interpreted not only in anatomical terms but also through the lens of circuit-level dynamics. Future work combining TUS with high-temporal-resolution methods or connectivity analyses could further elucidate the causal relationships between focal stimulation and distributed neural responses.

We have also expanded on our proposed explanation for the opposing effects of DBS and TUS found in this study. We now highlight that low frequency electrical stimulation has opposing effects to high frequency electrical stimulation with DBS. We include seminal references that demonstrate this and also highlight a recent publication in this journal (Darmani et al 2025) which reported an increase in beta power during low frequency TUS, an effect observed with low frequency DBS to the same network:

Regarding the polarity difference between the TUS and DBS results, perhaps the most likely explanation is that the specific DBS and TUS parameters employed had opposing physiological effects. The specific TUS parameters used here are thought to be excitatory^{23,61}. Conversely, though a simplification, high frequency DBS is thought of as functionally inhibitory⁷³. The behavioural effects we observed here are in keeping with the historical development of DBS for movement disorders, where high frequency DBS was observed to have the same clinical effect as lesions in non-human primates^{74,75}. Indeed whilst high frequency subthalamic DBS is known to improve symptoms of Parkinson's disease with an associated reduction of pathological beta power⁷⁶, low frequency DBS to the same region increases beta power⁷⁷ with a correlated worsening of symptoms⁷⁸. Similarly, low frequency TUS has recently been reported to increase beta power in the same target network⁷⁹. Nonetheless, the simplicity and directness of translation from excitatory and inhibitory physiological changes to behavioural facilitation and impairment, is unclear.

We have also outlined the possible reason for the DBS-Off condition being different to the sham condition:

Other factors might also be at play; namely the difference in baseline reward sensitivity seen between the healthy participants and the DBS cohort, all of whom had anorexia nervosa. It is known that reward sensitivity can be heightened in eating disorders^{81,82} and particularly anorexia nervosa⁸² compared to healthy controls and that the ability of an individual to learn from rewarding stimuli is reduced after DBS-NAcc⁸³. Therefore the DBS-NAcc results reported here are in keeping with those previously described in the literature, reported in anorexia nervosa for the first time.

5) The authors interpret non-reward outcomes as negative outcome, which is inaccurate. Negative outcome differs from no outcomes.

We thank the reviewer for raising this important distinction. We agree that non-reward is not equivalent to a negative outcome in the strictest sense; particularly when compared to explicit losses or punishments. We acknowledge that this is particularly important to bear in mind given the important role that dACC and adjacent perigenual ACC regions have in integrating information about the benefits of rewards with the potential costs of action in order to decide whether to engage with an opportunity (e.g. Amemori et al., 2012; Amemori et al., 2019; Amemori et al., 2024; Grohn et al., 2024; Khalighinejad et al., 2020). That said, our interpretation aligns with a large body of literature in reinforcement learning and behavioural neuroscience, in which reward omission is often perceived as *aversive* or *negative*, particularly in instrumental learning tasks where it signals an unfavorable outcome. In such contexts, the absence of expected reward carries negative value and can shape behaviour similarly to more explicit negative outcomes (e.g., Holroyd & Coles, 2002; Frank et al., 2007; Palminteri et al., 2015; Fouragnan et al., 2018).

We have clarified this in the revised manuscript to avoid any conflation between true losses and reward omission, and now refer to our outcomes as “non-reward” rather than “negative” throughout, while acknowledging the interpretative nuance involved. We believe this more accurately reflects the task design and the theoretical framework we build upon.

Extract from methods:

After making their decision, participants were shown either a tick to represent a reward on that trial or a cross to represent no reward. Participants were told that the number of rewarded trials would be counted across all the blocks and sessions, and that they would be awarded up to £10 as a performance bonus at the end of the study depending how well they performed during the task. **In the present task, outcomes were either reward or non-reward (i.e., reward omission); we use the term "non-reward" throughout to reflect the absence of positive feedback without implying an explicit negative or punishing outcome.**

6) The authors need to explore TUS effects on various behavioral elements more thoroughly, including learning, post-learning accuracy, perseveration errors, and post-error behavioral adjustments.

We thank the reviewer for this valuable suggestion. While we acknowledge that an expert such as the reviewer may be able to assimilate the results of multiple analyses, we are also painfully aware of the risks in overloading a manuscript with too many results which may be more than some readers can cope with. We believe the current manuscript already provides a comprehensive assessment of learning-related changes following TUS, including trial-by-trial learning curves and reinforcement learning models to estimate individual learning rates. These analyses revealed specific modulation of learning processes by TUS.

Nevertheless, in response to the reviewer’s helpful comment, we have now performed further analyses of post-learning accuracy, perseveration errors and post-error adjustment. While post-learning accuracy did not show significant effects of TUS (mixed-effects model [PostLearning_Acc ~ 1 + TUS_session + (1 | sub)]: $t_{75}=1.27$, $p=0.209$), a trend was observed for the NAcc (post-hoc t-test: NAcc-Sham: $p=0.07$). Perseveration errors did not differ significantly across conditions (mixed-effects model [Perf_Err ~ 1 + TUS_session + (1 | sub)]:

$t_{75}=0.363$, $p=0.717$) and post-error adjustment was also non significantly different across conditions (mixed-effects model [Post_Err ~ 1 + TUS_session + (1 | sub)]: $t_{75}=0.165$, $p=0.869$).

We agree that these additional behavioral markers are important avenues for future investigation, and this has now been acknowledged in the revised results section.

Extract from the results:

We also examined post-learning accuracy, perseveration errors, and post-error adjustments. These exploratory analyses did not reveal statistically significant differences between conditions. Specifically, post-learning accuracy showed no significant main effect of TUS (mixed-effects model [PostLearning_Acc ~ 1 + TUS_session + (1 | sub)]: $t_{75} = 1.27$, $p = 0.209$), though we observed a trend toward an effect in the NAcc condition (post-hoc t-test, NAcc vs. Sham: $p = 0.07$). Perseveration errors also did not differ significantly between TUS conditions (mixed-effects model [Perf_Err ~ 1 + TUS_session + (1 | sub)]: $t_{75} = 0.363$, $p = 0.717$), nor did post-error adjustment (mixed-effects model [Post_Err ~ 1 + TUS_session + (1 | sub)]: $t_{75} = 0.165$, $p = 0.869$). These analyses are reported for completeness and to guide future investigations.

7. As the study uses three stimuli, two with a 30% reward probability, in trials with two low-probability stimuli, there might be more ACC activation (as also shown by Hayden and Platt). I believe those trials need to be evaluated separately for the effect of TUS on the ACC, especially for BOLD signals.

We agree with the reviewer that these trials are potentially interesting for understanding dACC. Previous investigations of activity in both human dACC (Boorman et al., 2013) and macaque dACC (Fouragnan et al., 2019) have suggested that dACC reflects the value of alternative counterfactual choices to which an individual might switch on a future trial. Under normal circumstances, the likelihood of switching to the alternative choice is a function of the alternative option's value but this is no longer the case after TUS-induced disruption of dACC (Fouragnan et al., 2019). Thus in a situation in which both options that are available are low in value, we might expect that there would be some tendency to change behaviour as typically happens when, on average, the value of options is low (Wittmann et al., 2020; Priestley et al., in press). Such tracking of average value and its translation into a change in behaviour has been linked to the dorsal raphe nucleus (Wittmann et al., 2020; Priestley et al., in press). We might, however, expect this process to be held in check by a dACC mechanism that also holds precise evaluations of specific alternative choice options (as opposed to just average evaluations of all choices in an environment). Altering such dACC activity might, therefore, disrupt knowledge that the specific alternative that is available is not any higher in value than the choice taken (on trials where both options have a 30% chance of reward) but leave intact the general sense of average reward in the environment being low and that there is a need for a change in behaviour that is linked to the dorsal raphe nucleus. If this were to happen then we would expect animals to switch more frequently on trials with two low reward (30%) options after dACC TUS. This is, indeed, precisely what happens.

Suppl. Figure 1. Effect of dACC stimulation on switching behaviour after losses in low-probability (LP) trials. **Left:** Tendency to switch after losses in LP trials across dACC, NAcc, and Sham TUS conditions. TUS of dACC significantly increases switch behaviour compared to Sham ($p < 0.01$), while NAcc showed no significant effect. **Middle:** Time course of the effect of NAcc stimulation relative to Sham across four post-TUS testing blocks. No windows were significantly different than 0. **Right:** Time course of the effect of dACC stimulation relative to Sham across four post-TUS testing blocks. A trend-level increase in switch behaviour was observed approximately 35 minutes post-stimulation ($p = 0.06$), suggesting a temporally specific modulation.

We have now added this in the supplementary material. However, we attempted to identify neural correlates of these signals that might differ between TUS and Sham sessions, but found no significant effects. We believe this may be due to the analyses being underpowered.

8. The discussion is very brief and does not integrate the findings of the paper with the previous literature. The discussion needs to be better put together and discuss the connection of this paper to previous studies.

We have tried to revise the Discussion to provide a better summary of our findings and the reviewer's comments but there are also limits in the number of words that we can use. The revised Discussion now says:

This study aimed to capitalize on the high spatial resolution of TUS and its capacity to reach deep regions in the brain to target the NAcc in humans in the context of probabilistic reversal learning. A total of 26 healthy participants were enrolled in a within-subject repeated measures design experiment involving repetitive TUS and subsequent fMRI. After the application of 5Hz patterned TUS for 80 seconds in a counterbalanced fashion to the NAcc (TUS-NAcc), the dACC (TUS-dACC), or no sonication (Sham), participants performed a probabilistic reversal learning task in the MRI scanner which started on average 15min post sonication, when any potential auditory or somatosensory effects of stimulation were dissipated. We used both direct measures of reward sensitivity and model-based estimates of the expected value associated with each potential choice stimulus. The models were also used to examine prediction errors when participants received feedback to indicate if the choice was rewarded or unrewarded in analyses of both behaviour and neural activity. **We then compared these results with those of electrical deep brain stimulation to the NAcc (DBS-NAcc) in a rare cohort of patients with electrodes in the NAcc.**

With careful individualised TUS planning, using an estimate of each participant's skull image to achieve an optimised trajectory, it was possible to show that TUS-NAcc has neural effects that are most prominent in the region stimulated and that they are associated with changes in indices of behaviour that are similar to those emphasized in previous NAcc lesion studies^{43–46} but also in DBS-NAcc as observed in this study. Indeed, we found significant alterations in reward-related behaviours, including alterations in the tendency to adopt a win–stay strategy, and a changed learning curve for the rewarding option.

While TUS offers high spatial precision, its effects can extend beyond the immediate target due to both anatomical proximity and the broader functional connectivity of the stimulated region. Although the protocol was carefully optimized for focal delivery, and behavioural effects were limited to one stimulation site, whole-brain analyses revealed more distributed neural activity when directly comparing the two active TUS conditions. Importantly, these effects appeared more spatially confined

when each condition was compared to Sham, suggesting that the broader patterns reflect differential engagement of distinct networks rather than nonspecific or global activation.

This distinction is critical for interpreting the specificity of TUS effects. Localized behavioural outcomes, when paired with distributed neural changes, point toward functionally specific modulation within a larger interacting system. Indeed, evidence from the DBS literature suggests that stimulation of a single target can modulate activity of distinct neural networks in opposing directions, with both spatially⁶⁰ and temporally distinct dynamics^{61,62}. Furthermore, small variations in electrode position can account for widespread differences in network engagement across a variety of neuroanatomical targets^{63–66}. Therefore, while the spatial targeting of TUS remains a key strength, its influence should be interpreted not only in anatomical terms but also through the lens of circuit-level dynamics. Future work combining TUS with high-temporal-resolution methods or connectivity analyses could further elucidate the causal relationships between focal stimulation and distributed neural responses.

Importantly, however, while the impact of TUS to any brain region is likely to be mediated through the connectional network that that area has with the rest of the brain, it is equally important to remember that the connectional network of each area is unique⁶⁷. This means that while two areas, A and B, might share connections with one another, and the effect of stimulation to either area might partly be mediated by a change in activity induced in the other area, it is also the case that areas A and B will always each have distinct aspects to their connectional networks; some aspects of area A's and area B's connectional networks will be distinct from one another. This point was underlined in the current study when the effects of TUS to a distinct brain region, dACC, was studied. Although there are some similarities in the activity patterns found in dACC and NAcc, as well as in other areas that project to both areas such as the dopaminergic midbrain, there are also important differences⁵⁰. When TUS was applied to dACC in the current study, it did not change the aspects of reversal task performance that were affected by NAcc TUS even though previous studies examining the effect of lesions, microstimulation, or TUS to dACC and adjacent perigenual anterior cingulate cortex have demonstrated other alterations in behaviour such as changes in the ability to track the value of counterfactual choices – choices that were not taken on the current trial but to which the animal might switch in the future¹⁷ – and in learning from stochastic rewards and errors when an extended history of outcomes⁶⁸ or uncertainty⁴⁹ must be taken into account, and changes in cost-benefit integration and motivation for task engagement^{14,51,69–71} even though some of these effects are partly mediated via striatal regions adjacent to, and linked to, the NAcc^{70,71}.

Regarding the polarity difference between the TUS and DBS results, perhaps the most likely explanation is that the specific DBS and TUS parameters employed had opposing physiological effects. The specific TUS parameters used here are thought to be excitatory^{23,72}. Conversely, though a simplification, high frequency DBS is thought of as functionally inhibitory⁷³. The behavioural effects we observed here are in keeping with the historical development of DBS for movement disorders, where high frequency DBS was observed to have the same clinical effect as lesions in non-human primates^{74,75}. Indeed, whilst high frequency subthalamic DBS is known to improve symptoms of Parkinson's disease with an associated reduction of pathological beta power⁷⁶, low frequency DBS to the same region increases beta power⁷⁷ with a correlated worsening of symptoms⁷⁸. Similarly, low frequency TUS has recently been reported to increase beta power in the same target region⁷⁹. Nonetheless, the simplicity and directness of translation from excitatory and inhibitory physiological changes to behavioural facilitation and impairment, is unclear. Other factors might also be at play; namely the difference in baseline reward sensitivity seen between the healthy participants and the DBS cohort, all of whom had anorexia nervosa. It is known that reward sensitivity can be heightened in eating disorders^{80,81} and particularly anorexia nervosa⁸² compared to healthy controls and that the ability of an individual to learn from rewarding stimuli is reduced after DBS-NAcc⁸³. Therefore, the DBS-NAcc results reported here are in keeping with those previously described in the literature, reported in anorexia nervosa for the first time. It may be possible to identify TUS parameters that determine whether TUS exerts enhancing or disruptive effects. However, not only might the effects depend on many features of the TUS (intensity, frequency, and patterning of stimulation) but they might also depend on the anatomical structure of the brain region investigated, and the baseline behavioural tendencies of participants. Given the great interest in the possibility of TUS-based therapies², such factors merit careful consideration before TUS is employed in patients whose conditions may include a range of changes in reward sensitivity such as anorexia, substance use, bipolar disorder, or depression.

One limitation of the current study is the use of a constant free-field intensity for all participants, which may not effectively induce significant biological effects for some participants. Future research could focus on developing individualized neuromodulation protocols to maintain a constant in situ intensity for each participant. Additionally, customizing TUS parameters based on an individual's baseline reward sensitivity could enhance efficacy. Such personalized approaches are important for both research and therapeutic applications. As the field advances and safety data accumulates, using intensities closer to those in animal models may also yield stronger TUS effects on cognition and brain activity. **Finally, another limitation of the current study is the use of unilateral stimulation, which, although guided by current safety considerations for novel deep brain targets, does not allow us to fully assess potential lateralized effects in behaviour or neural response.**

This study provides evidence that TUS provides a minimally invasive method that can, in humans, manipulate activity in a deep subcortical region to induce early phase neuroplasticity¹². Many deep subcortical regions and subdivisions play crucial and specific roles in regulating fundamental behaviours and cognitive functions⁸⁴. These regions are difficult to access using traditional neuromodulation techniques and testing their causal roles in cognition in humans remains largely unexplored. Therefore, this study opens a potentially new and large space in which to examine hypotheses about human brain activity and its relationship with behaviours, with important lessons for future studies in neuropsychiatric conditions.

Minor comments:

The authors should report on reaction times for different conditions, as well as trials with and without high probability reward stimuli.

We thank the reviewer for this helpful suggestion. In response, we have now included analyses of reaction times across the different experimental conditions, as well as for trials involving high- versus low-probability reward stimuli. These results are now reported in the Supplementary Material. Briefly, we did not observe significant differences in reaction times across conditions or between trial types, although there was a trend toward slower responses in the NAcc condition relative to sham in the condition when the low probabilities outcomes were presented together. The reported p-values reflect the post hoc t-tests comparisons or tests against 0.

Suppl. Fig. 1. Reaction time analyses across stimulation conditions and trial types. Top row: Mean reaction times following TUS-dACC, TUS-NAcc, and Sham for all trials (left), HP trials (middle; trials with one high-probability and one low-probability option), and LP trials (right; trials with two low-probability options). **Middle row:** Reaction times across the four post-TUS blocks (~15, 28, 35, and 48 minutes) for each trial type and stimulation condition. **Bottom row:** Reaction times for rewarded trials only, split by trial type and time. No significant differences were observed across stimulation conditions, time blocks, or trial types, indicating that TUS did not affect motor response latency.

The text, in general, needs more edits, including technical ones. For example, on line 199: “we fitted the behavioural data to RL models and”; it should be the other way around (fitting models to the data).

We thank the reviewer for pointing out this error. We agree that the phrasing was inaccurate, models are, of course, fitted to the data, not the other way around. This has now been corrected in the revised manuscript, along with additional edits to improve clarity and technical accuracy throughout the text.

It is unclear whether each session involved only 3 stimuli or 12. If 12, then clarify on what is a reversal in this context, as it is not currently clear in the methods section.

This has now been clarified in the methods section:

Healthy participants performed a probabilistic reversal learning task during four blocks of MR acquisitions (two of which were fMRI acquisitions). The task consisted of two runs of 100 trials each (presented during the fMRI scans) and two runs of 60 trials each (no fMRI), giving a total of 320 trials across four blocks. **Three cues were presented per block, resulting in a total of 12 cues across the experiment** (adapted from ³⁶). Additional stimuli included a tick to represent a reward, a cross to represent no reward, and a fixation cross.

Potential off-target effects of TUS on other brain regions are not discussed despite broad neural activity changes noted.

The reviewer is correct in highlighting the importance of considering potential off-target effects of TUS, particularly in light of the broad neural activity changes observed in some contrasts. While TUS is designed to be spatially focused, it may still influence functionally connected regions through network-level interactions or, to a lesser extent, through physical spread of acoustic energy, especially in deep or densely interconnected areas.

In our study, broader activity changes were primarily observed when directly comparing the two active TUS conditions (TUS-dACC vs. TUS-NAcc), rather than when either was compared to Sham. This suggests that these differences reflect distinct patterns of engagement between the targeted regions, rather than nonspecific or widespread activation.

Importantly, the behavioral effects we observed were more circumscribed, emerging selectively in the TUS-NAcc condition and specifically affecting reward-guided learning processes, with no comparable changes in the TUS-dACC or Sham conditions. This dissociation between targeted behavioral effects and broader neural contrasts reinforces the notion that TUS can produce functionally specific outcomes, even in the context of distributed neural networks. Nonetheless, these findings underscore the importance of ongoing efforts to characterize the spatial extent and network-level consequences of TUS in both humans and animal models.

We have now added a section to the Discussion to highlight this important point, which we agree warrants explicit consideration.

Extract from the discussion:

While TUS offers high spatial precision, its effects can extend beyond the immediate target due to both anatomical proximity and the broader functional connectivity of the stimulated region. Although the protocol was carefully optimized for focal delivery, and behavioural effects were limited to one stimulation site, whole-brain analyses revealed more distributed neural activity when directly comparing the two active TUS conditions. Importantly, these effects appeared more spatially confined when each condition was compared to Sham, suggesting that the broader patterns reflect differential engagement of distinct networks rather than nonspecific or global activation.

This distinction is critical for interpreting the specificity of TUS effects. Localized behavioural outcomes, when paired with distributed neural changes, point toward functionally specific modulation within a larger interacting system. Indeed, evidence from the DBS literature suggests that stimulation of a single target can modulate activity of distinct neural networks in opposing directions, with both spatially⁶⁰ and temporally distinct dynamics^{61,62}. Furthermore, small variations in electrode position can account for widespread differences in network engagement across a variety of neuroanatomical targets⁶³⁻⁶⁶. Therefore, while the spatial targeting of TUS remains a key strength, its influence should be interpreted not only in anatomical terms but also through the lens of circuit-level dynamics. Future

work combining TUS with high-temporal-resolution methods or connectivity analyses could further elucidate the causal relationships between focal stimulation and distributed neural responses.

Importantly, however, while the impact of TUS to any brain region is likely to be mediated through the connectional network that that area has with the rest of the brain, it is equally important to remember that the connectional network of each area is unique⁶⁷. This means that while two areas, A and B, might share connections with one another, and the effect of stimulation to either area might partly be mediated by a change in activity induced in the other area, it is also the case that areas A and B will always each have distinct aspects to their connectional networks; some aspects of area A's and area B's connectional networks will be distinct from one another. This point was underlined in the current study when the effects of TUS to a distinct brain region, dACC, was studied. Although there are some similarities in the activity patterns found in dACC and NAcc, as well as in other areas that project to both areas such as the dopaminergic midbrain, there are also important differences⁵⁰. When TUS was applied to dACC in the current study, it did not change the aspects of reversal task performance that were affected by NAcc TUS even though previous studies examining the effect of lesions, microstimulation, or TUS to dACC and adjacent perigenual anterior cingulate cortex have demonstrated other alterations in behaviour such as changes in the ability to track the value of counterfactual choices – choices that were not taken on the current trial but to which the animal might switch in the future¹⁷ – and in learning from stochastic rewards and errors when an extended history of outcomes⁶⁸ or uncertainty⁴⁹ must be taken into account, and changes in cost-benefit integration and motivation for task engagement^{14,51,69–71} even though some of these effects are partly mediated via striatal regions adjacent to, and linked to, the NAcc^{70,71}.

In general, the text should clarify the RL model variables and their distinctions more effectively. It is unclear whether participants were the same or different across various conditions. Should be explained.

We thank the reviewer for this helpful comment. We have clarified the distinction between the reinforcement learning model variables (e.g., learning rates for rewarded vs. non-rewarded outcomes, prediction errors, and value estimates) and have explained how they were used in both behavioural and fMRI analyses. Specifically, we now describe in more detail how expected value and prediction error were computed and how they differ from directly observed behaviour. Furthermore, we have clarified in the Methods section that all participants completed all three TUS conditions (TUS-NAcc, TUS-dACC, and Sham) in a within-subjects counterbalanced design, ensuring that individual variability was controlled across conditions.

One of the methods section now reads:

The RL model estimated expected value (V), prediction error (δ), and separate learning rates (α^+ for reward, α^- for non-reward) for each participant, block, and TUS condition. While expected value reflects the ongoing belief about how rewarding each option is, prediction error reflects the mismatch between expected and actual outcomes, and drives updating of value estimates. These model-derived variables were used to examine latent learning processes and were analysed separately from directly observed behaviours, such as win-stay strategies.

Estimates from these models were used in both behavioural regressions and as parametric modulators in fMRI analyses, enabling us to dissociate stimulus-response learning from value-based decision signals.

Reviewer #2 (Remarks to the Author):

This is a beautifully written manuscript about a careful and well thought out set of experiments to demonstrate changes in behavior when sonicating the Nucleus Accumbens, as compared to control locations. It is an important study that should move the field forward. I have only a few comments that would provide clarity to the manuscript.

We are very grateful for the generous feedback. We deeply appreciate the comments about the quality of the writing and the design of the study. It is especially meaningful to us that the work is seen as a valuable contribution to the field, and we are glad that the manuscript's clarity and structure supported the reading of the results.

We also express our thanks for the suggestions aimed at further improving the manuscript. Detailed responses to each comment are provided below. We have carefully considered all feedback and revised the manuscript accordingly.

In Fig 1f, it is not clear what Mechanical Index is here. It is PNP derated by 0.3dB/cm/MHz and divided by the $\sqrt{f_0}$? Because this is the definition of mechanical index. Is it the PNP derated by some other derating factor for the skull and divided by the $\sqrt{f_0}$? This is then MI_{tc}. If it is MI_{tc}, then it should be labeled MI_{tc}.

We thank the reviewer for this helpful and entirely valid observation. The reviewer is correct that the Mechanical Index (MI) in Fig. 1f refers to the *in-situ* estimate, calculated using pressure values from transcranial simulations that account for skull attenuation. As such, this is not the free-field MI but rather the transcranial Mechanical Index (MI_{tc}), which uses derated PNP based on the individual acoustic simulations and is divided by $\sqrt{f_0}$. We have now updated the figure label and corresponding text in the figure legend and main manuscript to clearly reflect this, using the correct terminology "MI_{tc}" throughout. We agree that this clarification is important for accuracy and transparency, and we are grateful for the reviewer's attention to this detail.

Presumably, ISPPA at the focus and MI_{tc} would be related to each other. So, the maximum ISPPA at about 13 W/cm² would be related to the maximum MI_{tc}, which should be .91, but the figure has a maximum of .95. Can authors please explain.

We apologise for the confusion. Fig. 1f reported the MI_{tc} at the maximum peak in the head and the I_{SPPA} at the target coordinates inside the brain (which is the intended coordinate and not the max peak in the volume of the target), to account for (and as best proxy for) safety and efficacy, respectively. The maximum peak in the head is usually in the skull or nearby, not at the target. Therefore, for most individuals, the maximum MI_{tc} and I_{SPPA} at the target are not related. For simplicity, we report all single values in the supplementary material. We have revised the figure accordingly:

Figure 1. [...] Average and standard error of transcranial mechanical index at the maximum peak in the head, usually in the skull (left panel), I_{SPPA} at the intended brain target coordinate (middle panel), and occurrence of side effects after TUS-fMRI of the two TUS targets (right panel). Data from each individual participant are presented as small black circles.

In addition, while verifying the values in Supp. Tables 1 and 2, we found an error in some of the calculations of the $M_{I_{tc}}$ and I_{SPPA} which we have now corrected. The supplementary tables now read as:

Supplementary Table 1 (NAcc)

Suppl. Table 1. Acoustic simulation parameters and output for all study participants in the Nacc condition. The table reports the pressure values at the spatial-peak (Max Pressure), corresponding transcranial mechanical index ($M_{I_{tc}}$) and spatial-peak pulse-average intensity (I_{SPPA}), along with the maximum temperature rise in the head, to assess safety. To evaluate stimulation efficacy, the in situ estimate for the pressure amplitude at the target, I_{sppa} at the target and the size of the -6dB focal volume are reported. Pressure values are given in megapascals (MPa)

Focus depth	Max Pressure (MPa)	Mltc	Isppa (W/cm2)	Max. temp. rise	Pressure at target (MPa)	Isppa at target (W/cm2)	-6dB focal volume (mm3)
82	0.44	0.63	6.54	3.37	0.32	3.43	561.13
71	0.64	0.9	13.46	2.14	0.43	6.06	257.75
74	0.66	0.94	14.62	2.31	0.3	2.98	234.75
74	0.66	0.93	14.41	2.7	0.46	6.99	283.63
82	0.62	0.88	12.79	2.39	0.56	10.56	458.13
82	0.61	0.87	12.53	4.59	0.48	7.61	327
75	0.62	0.87	12.76	3.07	0.51	8.71	281.75
77	0.61	0.86	12.25	3.61	0.55	10.18	332
80	0.63	0.89	13.17	3.05	0.46	7.04	303.63
74	0.65	0.92	14.21	2.83	0.19	1.23	216
76	0.64	0.9	13.64	3.1	0.55	10	255.5
77	0.52	0.74	9.11	3.13	0.4	5.31	490.25
69	0.66	0.94	14.61	1.34	0.53	9.42	178.5
74	0.53	0.75	9.25	2.46	0.35	4.2	406.25
74	0.59	0.84	11.76	2.11	0.29	2.79	367.63
73	0.63	0.9	13.42	2.59	0.54	9.89	282.63
72	0.6	0.84	11.82	2.19	0.5	8.29	329
74	0.59	0.83	11.51	1.35	0.45	6.81	379.38
80	0.56	0.8	10.56	1.89	0.25	2.04	450.38
82	0.63	0.89	13.07	3.88	0.56	10.64	420.38
74	0.64	0.91	13.86	1.71	0.37	4.52	200.38
79	0.67	0.94	14.82	3.27	0.63	13.41	351.75
76	0.57	0.8	10.7	3.04	0.52	9.18	393.25
73	0.64	0.91	13.81	1.71	0.32	3.46	211
79	0.61	0.86	12.44	2.5	0.41	5.53	358.63
75	0.68	0.95	15.19	1.82	0.53	9.19	197.25

Supplementary Table 2 (dACC)

Suppl. Table 2. Acoustic simulation parameters and output for all study participants in the dACC condition. The table reports the pressure values at the spatial-peak (Max Pressure), corresponding transcranial mechanical index (Mltc) and spatial-peak pulse-average intensity (ISPPA), along with the maximum temperature rise in the head, to assess safety. To evaluate stimulation efficacy, the in situ estimate for the pressure amplitude at the target, Isppa at the target and the size of the -6dB focal volume are reported. Pressure values are given in megapascals (MPa)

Focus depth	Max Pressure (MPa)	Mltc	Isppa (W/cm2)	Max. temp. rise	Pressure at target (MPa)	Isppa at target (W/cm2)	-6dB focal volume (mm3)
64	0.64	0.9	13.59	1.95	0.59	11.6	175.75
57	0.64	0.9	13.61	0.93	0.52	9.01	113.63
57	0.52	0.73	8.86	0.94	0.41	5.6	210.38
61	0.61	0.86	12.31	1.09	0.49	8	175.13
57	0.65	0.92	14.06	0.85	0.52	9.01	132.38
58	0.57	0.81	10.99	1.01	0.39	5.07	166.13
57	0.57	0.8	10.78	0.9	0.37	4.56	167.63
54	0.61	0.86	12.28	0.73	0.6	12	115.13
58	0.47	0.66	7.21	1.21	0.43	6.16	278.38
57	0.51	0.72	8.67	1.01	0.37	4.56	201.88
61	0.55	0.78	10.26	1.29	0.42	5.88	207.5
58	0.53	0.75	9.41	0.88	0.41	5.6	207
57	0.62	0.87	12.74	0.81	0.53	9.36	131.88
61	0.65	0.92	14.03	1.28	0.52	9.01	144
61	0.64	0.9	13.47	1.39	0.43	6.16	158.75
60	0.62	0.88	12.98	1.16	0.51	8.67	165.13
53	0.55	0.78	10.09	0.95	0.43	6.16	163.63
59	0.61	0.86	12.42	1	0.4	5.33	157.25
62	0.45	0.63	6.67	1.16	0.35	4.08	92.63
62	0.64	0.9	13.54	1.35	0.47	7.36	159.38
65	0.49	0.69	7.97	1.63	0.39	5.07	128.25
69	0.61	0.87	12.5	1.46	0.48	7.68	84.25
63	0.56	0.8	10.58	1.2	0.54	9.72	196.13
60	0.65	0.93	14.29	1.09	0.56	10.45	141.63
58	0.54	0.76	9.57	1.21	0.38	4.81	196.13
64	0.52	0.73	8.98	1.57	0.48	7.68	262.75

Supp Table 1 - are columns 2-4 in free field? No, the methods says that the free field value was kept at 30 W/cm2. So what are these three columns? Same for supplementary table 2.

The reported values for Pressure and I_{SPPA} are all derated values based on the individual acoustic simulations. Following the recommendations from the ITRUSST consensus on standardised reporting for TUS (Martin et al., 2024), we report both the *in situ* estimate for the spatial-peak pressure amplitude (maximum anywhere in the head), and the *in situ* estimate for the pressure amplitude at the target (intended target coordinate). We have now clarified the Table header and legend as above.

page 16, the transducer is referred to as a bespoke transducer. According to miriam webbster, bespoke means custom made. Is this transducer custom made? If it is custom made, there should be more details about it.

We thank the reviewer for this careful reading and for the opportunity to clarify the use of the term “bespoke.” The reviewer is absolutely correct that “bespoke” typically implies a custom-made device. In this case, the transducer was indeed custom specified at the time of purchase.

This particular unit was one of the first transducers ever purchased from Sonic Concepts for use in neuromodulation (in 2019), prior to the release of their current CTX-500 series. At the time, rather than offering a fixed steering range, Sonic Concepts invited user input regarding key design specifications. We were specifically asked to define the desired steering range, which we selected to accommodate deeper targets such as the human NAcc. As a result, our transducer differs from more recent commercial CTX-500 units, which have a standardised and shallower steering range.

To reflect this, we have added further details to the Methods section describing the transducer, including its customisable aspects and its historical context. We hope this additional information provides greater clarity and appropriately supports the use of the term “bespoke” in this instance.

A bespoke CTX-500 NeuroFUS TPO system (Brainbox Ltd., Cardiff, UK) with a four-element annular transducer (diameter=64 mm, central frequency=500 kHz, and steering range between 27.3 and 82.6mm) was used to deliver 5 Hz pulse repetition frequency repetitive TUS (pulse duration=20 ms, pulse repetition interval=200 ms, total duration=80 s, total number of pulses=400). *The term **bespoke** reflects the custom specification of the transducer at the time of purchase in 2019, prior to the release of the standardized CTX-500 series. At that time, Sonic Concepts invited user-defined design parameters, including steering range. We selected a steering range (27.3–82.6 mm) tailored for targeting deep structures such as the human NAcc. As such, this transducer differs from later commercial CTX-500 units, which feature shallower, fixed ranges.* This steering range ensured that it was possible to reach the NAcc in humans. The target free-field spatial-peak pulse-average intensity (ISPPA) was kept constant at ~30 W/cm² for all participants, which is the intensity before going through the skull *and the soft tissue*. Transcranial acoustic and thermal simulations (see “Acoustic simulations” section for details) were performed both during planning and after each session to confirm that transcranial intensities remained within the limits of the ITRUSST safety guidelines for TUS ²⁸.

page 16, line 442, before going through the skull and presumably also the soft tissues.

Great observation. We've updated the sentence to read as follows:

The target free-field spatial-peak pulse-average intensity (ISPPA) was kept constant at ~30 W/cm² for all participants, which is the intensity before going through the skull *and the soft tissues*.

Reviewer #3 (Remarks to the Author):

This study applied 80 s of low-intensity pulsed transcranial ultrasound to NAcc and dACC of 26 human subjects engaged in a probabilistic reward reversal task. In this task, the subjects had to learn which of three symbols is associated with the highest reward, before the reward contingencies switched. There was a selective enhancement in parametric BOLD for both NAcc and dACC, relatively to sham.

The effects on behavior were assessed post-sonication, to eliminate potential artifacts that can be associated with the ultrasound.

There were substantial effects of NAcc stimulation (but not dACC stimulation) on the tendency to stay following a reward. The probability of choosing the high-probability reward symbol was increased following NAcc stimulation. There was also an appreciable effect on the learning rate following a reward. The first two effects (tendency to stay following a reward and probability of choosing the high-probability reward symbol) were also observed in participants receiving NAcc DBS, although the ultrasound and DBS effects were of opposite polarity.

I find the study well executed. There is a sham condition, a control location (dACC), and the effects of ultrasound on behavior are evaluated first following the ultrasound offset, thus eliminating potential artifacts.

The demonstration of target engagement using fMRI BOLD is equally important.

The study is also well written.

The finding that effects of deep brain ultrasound stimulation on learning are observed even under relatively low (and safe) exposure levels qualifies the study, in my opinion, to be published.

Thank you very much for your thoughtful and positive feedback. We are especially pleased that you appreciated the methodological aspects of the study, including the use of a sham condition, a control site (dACC), and the post-sonication behavioural assessments to avoid potential sound artefacts. We also agree that demonstrating neural target engagement through fMRI was a crucial step towards establishing the specificity and validity of the TUS effects. We are also grateful for your kind words regarding the clarity and writing of the manuscript.

I was only puzzled by the ultrasound and the DBS showing opposite effects. It is nonetheless possible, as the authors describe, that the two modalities do provide effects of opposite polarity under the stimuli used.

As the reviewer noted, the enhancement in reward-guided behaviours following TUS-NAcc particularly increased win–stay behaviour and selection of the high-probability option, aligns with prior lesion and stimulation findings in non-human primates. Your observation regarding the parallel effects seen with DBS-NAcc, but in the opposite direction, is especially insightful.

This polarity difference is indeed striking. While both TUS and DBS influenced the same behavioural markers of reward sensitivity, TUS appeared to enhance them, whereas DBS reduced them. We interpret this contrast as likely reflecting fundamental differences in their mechanisms of action. High-frequency DBS is often considered functionally inhibitory, similar to lesioning, while the TUS protocol used here, being low intensity and applied offline, is more likely to be excitatory and potentially engage plasticity-like processes.

At the same time, individual context may matter. The DBS participants had elevated reward sensitivity at baseline (OFF state), and the observed reduction with DBS-ON may reflect a normalising effect rather than simple inhibition. In contrast, TUS was applied to healthy participants, potentially enhancing reward processing beyond typical levels.

We now expand on this in the discussion section of the manuscript. Extract from discussion:

Regarding the polarity difference between the TUS and DBS results, perhaps the most likely explanation is that the specific DBS and TUS parameters employed had opposing physiological effects. The specific TUS parameters used here are thought to be excitatory^{23,72}. Conversely, though a simplification, high frequency DBS is thought of as functionally inhibitory⁷³. The behavioural effects we observed here are in keeping with the historical development of DBS for movement disorders, where high frequency DBS was observed to have the same clinical effect as lesions in non-human primates^{74,75}. Indeed, whilst high frequency subthalamic DBS is known to improve symptoms of Parkinson's disease with an associated reduction of pathological beta power⁷⁶, low frequency DBS to the same region increases beta power⁷⁷ with a correlated worsening of symptoms⁷⁸. Similarly, low frequency TUS has recently been reported to increase beta power in the same target region⁷⁹. Nonetheless, the simplicity and directness of translation from excitatory and inhibitory physiological changes to behavioural facilitation and impairment, is unclear. Other factors might also be at play; namely the difference in baseline reward sensitivity seen between the healthy participants and the DBS cohort, all of whom had anorexia nervosa. It is known that reward sensitivity can be heightened in eating disorders^{80,81} and particularly anorexia nervosa⁸² compared to healthy controls and that the ability of an individual to learn from rewarding stimuli is reduced after DBS-NAcc⁸³. Therefore, the DBS-NAcc results reported here are in keeping with those previously described in the literature, reported in anorexia nervosa for the first time. It may be possible to identify TUS parameters that determine whether TUS exerts enhancing or disruptive effects. However, not only might the effects depend on many features of the TUS (intensity, frequency, and patterning of stimulation) but they might also depend on the anatomical structure of the brain region investigated, and the baseline behavioural tendencies of participants. Given the great interest in the possibility of TUS-based therapies², such factors merit careful consideration before TUS is employed in patients whose conditions may include a range of changes in reward sensitivity such as anorexia, substance use, bipolar disorder, or depression.

Altogether, we agree that this divergence between modalities deserves further investigation. It highlights how the behavioural outcomes of neuromodulation depend on the interaction between stimulation parameters, brain region, and individual baseline function.

You will find below the reviewers' comments numbered and in normal font in black. Responses are in blue.

REVIEWERS' COMMENTS

Reviewer #1 (Remarks to the Author):

Authors have done a great job in addressing my comments. I have no further comment.

We thank the reviewer.

Reviewer #2 (Remarks to the Author):

Thank you for the thoughtful responses. Just one comment

Figure 1f still does not clarify whether it is MI or MI_{tc}. The response letter suggests it is MI_{tc}, but neither the graph label, nor the figure legend say this. In addition, the figure in the manuscript was not updated, as it does not match the figure in the response letter. But, even that figure does not clarify it is MI_{tc} in the figure label or in the caption.

We thank the reviewer for this comment. We are not entirely sure why this appeared unclear, as both the figure and caption clearly indicate MI_{tc}. We have double checked and confirmed that it is present, and we hope the reviewer will be satisfied with the edits.

Reviewer #3 (Remarks to the Author):

The authors have addressed my suggestion.

We thank the reviewer.